# An Explicit Meshless Point Collocation Solver for Incompressible Navier-Stokes Equations

**George C. Bourantas** [1,*], **Benjamin F. Zwick** [1], **Grand R. Joldes** [1], **Vassilios C. Loukopoulos** [2], **Angus C. R. Tavner** [1], **Adam Wittek** [1]  **and Karol Miller** [1]

[1] Intelligent Systems for Medicine Laboratory, The University of Western Australia, 35 Stirling Highway, Perth, WA 6009, Australia; benjamin.zwick@research.uwa.edu.au (B.F.Z.); grand.joldes@uwa.edu.au (G.R.J.); angus.tavner@uwa.edu.au (A.C.R.T.); adam.wittek@uwa.edu.au (A.W.); karol.miller@uwa.edu.au (K.M.)

[2] Department of Physics, University of Patras, Patras, 26500 Rion, Greece; vxloukop@physics.upatras.gr

[*] Correspondence: george.bourantas@uwa.edu.au

**Abstract:** We present a strong form, meshless point collocation explicit solver for the numerical solution of the transient, incompressible, viscous Navier-Stokes (N-S) equations in two dimensions. We numerically solve the governing flow equations in their stream function-vorticity formulation. We use a uniform Cartesian embedded grid to represent the flow domain. We discretize the governing equations using the Meshless Point Collocation (MPC) method. We compute the spatial derivatives that appear in the governing flow equations, using a novel interpolation meshless scheme, the Discretization Corrected Particle Strength Exchange (DC PSE). We verify the accuracy of the numerical scheme for commonly used benchmark problems including lid-driven cavity flow, flow over a backward-facing step and unbounded flow past a cylinder. We have examined the applicability of the proposed scheme by considering flow cases with complex geometries, such as flow in a duct with cylindrical obstacles, flow in a bifurcated geometry, and flow past complex-shaped obstacles. Our method offers high accuracy and excellent computational efficiency as demonstrated by the verification examples, while maintaining a stable time step comparable to that used in unconditionally stable implicit methods. We estimate the stable time step using the Gershgorin circle theorem. The stable time step can be increased through the increase of the support domain of the weight function used in the DC PSE method.

**Keywords:** transient incompressible Navier-Stokes; meshless point collocation method; stream function-vorticity formulation; strong form; explicit time integration

## 1. Introduction

In this paper, we describe the development of a strong form meshless point collocation method for numerically solving the two-dimensional, non-stationary, incompressible Navier-Stokes (N-S) equations in their stream function-vorticity formulation. Our method relies on the ability of meshless methods to accurately compute spatial derivatives, and on their flexibility to solve partial differential equations (PDEs) in complex geometries. The proposed numerical method can efficiently handle uniform Cartesian point clouds embedded in a complex geometry, and/or irregular sets of nodes that directly discretize the flow domain. To compute the spatial derivatives that appear in the governing flow equations, we apply an interpolation meshless scheme, which computes spatial derivatives on both Cartesian embedded grids and irregular nodal distributions.

The mathematical formulation of the N-S equations, depending on the choice of dependent variables, can be classified as (i) primitive variables, (ii) stream function-vorticity, and (iii) velocity-vorticity. The majority of numerical methods are expressed in the primitive variables

(velocity-pressure) formulation [1–3]. Although the primitive variables formulation is widely employed, difficulties occur when dealing with pressure boundary conditions, and coupling of the velocity and pressure fields is challenging. The vorticity-vector-potential [4–6] and velocity-vorticity [7] formulations have a clear advantage over the primitive variables formulation, because pressure does not appear in the field equations, and the difficulty of determining the pressure boundary conditions in incompressible flows is avoided [8]. The extension to 3D flow is also straightforward, although the number of field variables increases from three to six. For 2D cases, stream function-vorticity is the preferable method for the numerical solution of the incompressible N-S equations, since the continuity equation (conservation of mass) is inherently fulfilled [4].

The non-stationary N-S equations can be numerically solved using explicit, explicit-implicit or implicit time integration schemes in the framework of finite element (FE), finite volume (FV), finite difference (FD) or spectral methods. Fletcher [9] provides an overview of finite element and finite difference techniques for incompressible fluid flow, while Gresho and Sani [10] review the field, focusing on finite elements. Standard texts on computational fluid dynamics (CFD) using finite difference and finite volume methods can be found in [11–13], and finite elements in [14–16].

Substantial effort has been invested in reducing or eliminating problems related to mesh generation. Attempts to eliminate the human and computational effort related to mesh generation have resulted in the development of meshless methods (MMs). Various MMs, developed for computational fluid dynamics (CFD), have been used to solve the incompressible N-S equations. Of particular interest is the Meshless Local Petrov-Galerkin (MLPG) method, which has been used to solve the N-S equations in their primitive variables [17], velocity-vorticity [18,19] and stream function-vorticity [20] formulations. The moving least squares (MLS) approximation method has been used to compute shape functions and derivatives. Additionally, radial basis functions (RBFs) are able to interpolate shape function and its derivatives [21]. The Local Boundary Integral Method (LBIE) has been successfully applied to solve incompressible, laminar flow cases [22], while the multi-region Local Boundary Integral Equation (LBIE) has been combined with the radial basis functions (RBF) interpolation method [23]. The mesh-free Local Radial Basis Function Collocation Method (LRBFCM) for transient heat transfer and fluid dynamics problems was presented in [24], while others [25,26] reported the use of an indirect/integrated radial-basis-function network (IRBFN) method to solve transient partial differential equations (PDEs) governing fluid flow problems. Velocity-vorticity flow equations were solved in [27–29] using a point collocation method, along with a velocity-correction method, while a meshless point collocation with both velocity and pressure correction methods [30] was applied in several fluid mechanics problems. Eulerian based meshless methods, in contrast to Lagrangian based meshless methods, such as Smoothed Particle Hydrodynamics (SPH), have been mainly applied to regular geometries (square, circles, tubes), where generating a computational grid is relatively straightforward.

In this work, we focus on the Eulerian formulation of the N-S equations. A meshless point collocation method is used for the numerical solution of the non-stationary, incompressible, laminar 2D N-S equations in their stream function-vorticity formulation in complex geometries. The stream function-vorticity formulation is an effective way to describe 2D incompressible flow, since the incompressibility constraint for the velocity is automatically satisfied and the pressure variable is eliminated. The method is generalizable to three dimensions via vector potential-vorticity formulation [4–6]. The vorticity boundary conditions are imposed in a local form. We treat the transient term using a Euler explicit time integration scheme. Since the explicit integration scheme is conditionally stable, we compute the critical time step that ensures stable numerical results using the Gershgorin circle theorem [31,32]. This circumvents the need to calculate eigenvalues, which would be time consuming. The requirement for a stable time-step is counterbalanced by the low computational cost of each step, since only one Poisson-type equation is solved for the stream function equation and the vorticity field is updated using the explicit solver (in three dimensions three Poisson-type equations need to be solved [4–6]). The proposed scheme applies to both uniform embedded and irregular nodal distributions and can be used with ease in complex geometries.

This paper is structured as follows: in Section 2, we (i) present the governing equations (ii) present a numerical scheme for the stream function-vorticity formulation based on the meshless point collocation method (iii) briefly describe the Discretization Correction Particle Strength Exchange (DC PSE) differentiation method and (iv) discuss issues related to the accuracy and the computational cost of the proposed scheme. In Section 3, we highlight the accuracy of the scheme through benchmark problems, while in Section 4, we demonstrate the accuracy, efficiency and the ease of use of the proposed scheme using numerical examples. Finally, in Section 5 we present our conclusions and preliminary results for 3D lid-driven cavity flow using the generalization of the proposed scheme via vorticity-vector potential formulation.

## 2. Methods Governing Equations and Solution Procedure

### 2.1. Stream Function-Vorticity Formulation of Navier-Stokes Equations

The governing equations for the non-stationary, viscous, laminar flow of an incompressible fluid are based on conservation of mass and momentum. The non-dimensional form of the stream function-vorticity ($\psi$-$\omega$) formulation is:

$$\frac{\partial \omega}{\partial t} + \frac{\partial \psi}{\partial y}\frac{\partial \omega}{\partial x} - \frac{\partial \psi}{\partial x}\frac{\partial \omega}{\partial y} = \frac{1}{Re}\nabla^2\omega \tag{1}$$

$$\nabla^2\psi = -\omega \tag{2}$$

where $Re = \frac{UL}{v}$ is the Reynolds number, with $U$ and $L$ being a characteristic velocity and length, respectively, and $v$ the kinematic viscosity of the fluid. Taking the spatial derivative of the stream function $\psi$ yields the velocity components $u$ and $v$, as follows:

$$u = \frac{\partial \psi}{\partial y} \tag{3}$$

$$v = -\frac{\partial \psi}{\partial x} \tag{4}$$

We require a solution to Equations (1) and (2), defined in the spatial domain $\Omega$ with boundary $\partial\Omega$, given the initial conditions at time $t = 0$

$$\boldsymbol{u} = \boldsymbol{u}_0 \tag{5}$$

$$\omega = \omega_0 = \nabla \times \boldsymbol{u}_0, \tag{6}$$

and boundary conditions

$$\boldsymbol{u} = \boldsymbol{u}_{\partial\Omega} \tag{7}$$

$$\omega = (\nabla \times \boldsymbol{u}_{\partial\Omega}) \cdot \hat{\boldsymbol{k}} \tag{8}$$

with $\hat{\boldsymbol{k}}$ being the unit vector in the direction of the $z$ axis of a three-dimensional Cartesian coordinate system. We solve the governing equations using an explicit Euler time integration scheme for the transient terms and a strong form meshless collocation method to discretize the flow equations.

### 2.2. Temporal Discretization Using Explicit Euler Method

We treat the viscous term explicitly [33] to obtain the following difference formula in time:

$$\frac{\omega^{n+1} - \omega^n}{\Delta t} + \frac{\partial \psi^n}{\partial y}\frac{\partial \omega^n}{\partial x} - \frac{\partial \psi^n}{\partial x}\frac{\partial \omega^n}{\partial y} = \frac{1}{Re}\nabla^2\omega^n. \tag{9}$$

$$\nabla^2\psi^{n+1} = -\omega^{n+1} \tag{10}$$

using the notation $\omega^{n+1} = \omega\left(t^{n+1}\right)$ and $\psi^{n+1} = \psi\left(t^{n+1}\right)$, where $t^{n+1} = t^n + \delta t$. The proposed scheme computes the stream function (along with velocity components) and vorticity field by applying a three-step marching algorithmic procedure:

Step 1　Update vorticity at the interior grid points

$$\omega^{n+1} = \omega^n - \Delta t \frac{\partial \psi^n}{\partial y} \frac{\partial \omega^n}{\partial x} + \Delta t \frac{\partial \psi^n}{\partial x} \frac{\partial \omega^n}{\partial y} + \Delta t \frac{1}{Re}\left(\frac{\partial^2 \omega^n}{\partial x^2} + \frac{\partial^2 \omega^n}{\partial y^2}\right) \tag{11}$$

Step 2　Compute stream function by solving Poisson type problem

$$\frac{\partial^2 \psi^{n+1}}{\partial x^2} + \frac{\partial^2 \psi^{n+1}}{\partial y^2} = -\omega^{n+1} \tag{12}$$

Step 3　Update velocity $\boldsymbol{u}^{(n+1)}$

$$u^{n+1} = \frac{\partial \psi^{n+1}}{\partial y} \tag{13}$$

$$v^{n+1} = -\frac{\partial \psi^{n+1}}{\partial x} \tag{14}$$

### *2.3. Discretization Corrected Particle Strength Exchange (DC PSE) Method*

The accuracy of the proposed scheme relies on the accurate computation of spatial derivatives for the unknown field functions in Equations (11)–(14). In the present study, we use the Discretization Corrected Particle Strength Exchange (DC PSE) method, which, to our knowledge, is one of the most accurate meshless methods for computing spatial derivatives. The DC PSE method originated as a Lagrangian particle-based numerical method [34] and is based on Particle Strength Exchange (PSE) operators. To solve PDEs using the DC PSE meshless method, the authors in [35] reformulated the Lagrangian DC PSE method to work in the Eulerian framework. For completion, the PSE operators and the DC PSE method are described below.

### 2.3.1. Particle Strength Exchange (PSE) Operators

The Particle Strength Exchange (PSE) method utilizes kernels to approximate differential operators that conserve particle strength in particle-particle interactions. The PSE method was proposed by Degond and Mas-Gallic [36] for diffusion and convection-diffusion problems. Eldredge et al. [37] then developed a framework for approximating arbitrary derivatives.

In general, a PSE operator $Q^\beta f(\boldsymbol{x})$ for approximating the derivative $D^\beta f(\boldsymbol{x})$ has the form

$$Q^\beta f(\boldsymbol{x}) = \frac{1}{\varepsilon^{|\beta|}} \int (f(\boldsymbol{y}) \mp f(\boldsymbol{x}))\eta_\varepsilon^\beta(\boldsymbol{x} - \boldsymbol{y})d\boldsymbol{y} \tag{15}$$

with $\eta_\varepsilon^\beta = \eta^\beta(x/\varepsilon)/\varepsilon^d$ a scaled kernel of radius $\varepsilon$, and $d$ the number of dimensions. The sign in Equation (15) is negative when $|\beta|$ is even and positive when $|\beta|$ is odd, with $\boldsymbol{\beta}$ a multi-index [34]. Now, let $\boldsymbol{\beta} = (\beta_1, \beta_2, \ldots, \beta_n)$, where $\beta_i$, $i = 1, 2, \ldots, n$ is a non-negative integer. Then the partial differential operator $D^\beta$ can be expressed as $D^\beta = \frac{\partial^{|\beta|}}{\partial x_1^{\beta_1} \partial x_2^{\beta_2} \ldots \partial x_n^{\beta_n}}$. The challenge is to find a kernel $\eta_\varepsilon^\beta$ that leads to good approximations for $D^\beta$. To find such kernels for arbitrary derivatives we adopt the idea in Eldredge et al. [37] and start from the Taylor expansion of a function $f(\boldsymbol{y})$ about a point $\boldsymbol{x}$:

$$f(\boldsymbol{y}) = f(\boldsymbol{x}) + \sum_{|a|=1}^{\infty} \frac{1}{a!}(\boldsymbol{y} - \boldsymbol{x})^a D^a f(\boldsymbol{x}). \tag{16}$$

Next, we subtract or add $f(x)$, depending on whether $|\beta|$ is odd or even, to both sides and convolute the equation with the unknown kernel $\eta_\varepsilon^\beta$. Considering Equation (15), this leads to:

$$Q^\beta f(x) = \frac{1}{\varepsilon^{|\beta|}} \sum_{|a|=1}^{\infty} \frac{1}{a!} D^a f(x) \int (y-x)^a \eta_\varepsilon^\beta(x-y) dy. \tag{17}$$

Introducing the continuous $\alpha$-moments

$$M_a = \int (x-y)^\alpha \eta^\beta(x-y) dy = \int z^\alpha \eta^\beta(z) dz \tag{18}$$

and isolating the derivatives $D^\beta$ on the right-hand side of Equation (17) we obtain:

$$Q^\beta f(x) = \frac{(-1)^{|\beta|}}{\beta!} M_\beta D^\beta f(x) + \sum_{\substack{|a|=1 \\ a \neq \beta}}^{\infty} \frac{(-1)^{|a|}}{a!} \varepsilon^{|a|-|\beta|} M_a D^a f(x). \tag{19}$$

Finally, to approximate $Q^\beta f(x)$ with order of accuracy $r$, the following set of conditions is imposed for the moments $M_a$:

$$M_a = \begin{cases} (-1)^{|\beta|}\beta! & a = \beta \\ 0, & a \neq \beta, \quad 1 \leq |a| \leq |\beta| + r - 1. \end{cases} \tag{20}$$

In addition, if we impose

$$\int |z|^{|\beta|+r} |\eta^\beta(z)| dz < \infty \tag{21}$$

the mollification error $\epsilon_\varepsilon(x) = D^\beta f(x)$ is bounded [34]. The procedure to construct a kernel that satisfies the conditions in Equation (16) has been described in [34]. Once the kernel is defined, the operator in Equation (17) can be discretized using a midpoint quadrature over the nodes as

$$Q_h^\beta f(x) = \frac{1}{\varepsilon^{|\beta|}} \sum_{p \in N(x)} \left( f(x_p) \mp f(x) \right) \eta_\varepsilon^\beta(x-x_p) V_p , \tag{22}$$

where $N(x)$ is the number of all nodes in a neighborhood around $x$, which can be defined by a cut-off radius $r_c$, usually chosen such that $N(x)$ coincides to a certain level of accuracy with the kernel support, and $V_p$ is the volume associated with each particle. Given such a discretization, the discretization error $\epsilon_h(x) = Q^\beta f(x) - Q_h^\beta f(x)$ is also bounded [34].

### 2.3.2. The Discretization Corrected PSE Operators

The Discretization Corrected PSE operators were introduced by Schrader et al. [34] to reduce the discretization error $\epsilon_h(x)$ in the PSE operator approximation. We can derive the DC PSE approximation from Equation (22), aiming of finding a kernel function which minimizes the difference between this discrete operator and the actual derivative. We can achieve this, by replacing each term $f(x_p)$ in Equation (22) with its Taylor expansion about $x$. This leads to the following expression for the derivative approximation:

$$Q_h^\beta f(x) = \frac{(-1)^{|\beta|}}{\beta!} Z_h^\beta D^\beta f(x) + \sum_{\substack{|a|=1 \\ a \neq \beta}}^{\infty} \frac{(-1)^{|a|}}{a!} \varepsilon^{|a|-|\beta|} Z_h^a D^a f(x) + r_0 \tag{23}$$

with

$$r_0 = \begin{cases} 0, & |\beta| \text{ even} \\ 2e^{-|\beta|}Z_h^0 f(\boldsymbol{x}) & |\beta| \text{ odd} \end{cases} \tag{24}$$

and the discrete moments defined as:

$$Z_h^a = \frac{1}{\varepsilon^d} \sum_{p \in N(\boldsymbol{x})} \left(\frac{\boldsymbol{x} - \boldsymbol{x}_p}{\varepsilon}\right)^a \eta^\beta\left(\frac{\boldsymbol{x} - \boldsymbol{x}_p}{\varepsilon}\right). \tag{25}$$

Therefore, the set of moment conditions becomes:

$$Z_h^a = \begin{cases} (-1)^{|\beta|}\beta! & a = \beta \\ 0 & a \neq \beta \qquad a_{min} \leq |a| \leq |\beta| + r - 1 \\ < \infty & \text{otherwise} \end{cases} \tag{26}$$

for all $|\beta| \neq 0$, where $a_{min}$ is one when $|\beta|$ is odd and zero when $|\beta|$ is even. The kernel $\eta^\beta$ is chosen as:

$$\eta^\beta(\boldsymbol{x}, \boldsymbol{z}) = \left(\sum_{|\gamma| = a_{min}}^{|\beta| + r - 1} a_\gamma(\boldsymbol{x}) \boldsymbol{z}^\gamma\right) e^{-|\boldsymbol{z}|^2} = P(\boldsymbol{x}, \boldsymbol{z})W(\boldsymbol{z}), \quad \boldsymbol{z} = \frac{\boldsymbol{x} - \boldsymbol{x}_p}{\varepsilon}. \tag{27}$$

The kernel function consists of a polynomial correction function $P(\boldsymbol{x}, \boldsymbol{z})$ and the weight function $W(\boldsymbol{z})$. Different types of weight functions can be applied with the possible choices described in [38].

The unknown coefficients $a_\gamma(\boldsymbol{x})$ are obtained by requiring the kernel given by Equation (28) to satisfy the conditions in Equation (27), resulting in the following linear system of equations:

$$\sum_{|\gamma| = a_{min}}^{|\beta| + r - 1} a_\gamma(\boldsymbol{x}) w_a^\gamma(\boldsymbol{x}) = \begin{cases} (-1)^{|\beta|}\beta! & a = \beta \\ 0 & a \neq \beta \end{cases}, \quad \forall\, a_{min} \leq |a| \leq |\beta| + r - 1 \tag{28}$$

with weights

$$w_a^\gamma(\boldsymbol{x}) = \frac{1}{\varepsilon^{|a+\gamma|+d}} \sum_{p \in N(\boldsymbol{x})} \left(\boldsymbol{x} - \boldsymbol{x}_p\right)^{a+\gamma} e^{-\left(\frac{|\boldsymbol{x} - \boldsymbol{x}_p|}{\varepsilon}\right)^2} \tag{29}$$

To compute the approximated derivative $Q_h^\beta f(\boldsymbol{x})$ at node $\boldsymbol{x}_p$, the coefficients are found by solving the linear system of Equation (28) for $\boldsymbol{x} = \boldsymbol{x}_p$. Given our choice of kernel function, the DC PSE derivative approximation becomes:

$$Q_h^\beta f(\boldsymbol{x}_p) = \frac{1}{\varepsilon(\boldsymbol{x}_p)^\beta} \sum_{\boldsymbol{x}_q \in N(\boldsymbol{x}_p)} \left(f(\boldsymbol{x}_q) \mp f(\boldsymbol{x}_p)\right) p\left(\frac{\boldsymbol{x} - \boldsymbol{x}_p}{\varepsilon(\boldsymbol{x}_p)}\right) a^T(\boldsymbol{x}_p) e^{-\left(\frac{|\boldsymbol{x}_p - \boldsymbol{x}_q|}{\varepsilon(\boldsymbol{x}_p)}\right)^2} \tag{30}$$

where $p(\boldsymbol{x}) = [p_1(\boldsymbol{x}), p_2(\boldsymbol{x}), p_l(\boldsymbol{x})]$ and $a(\boldsymbol{x})$ are the vectors of terms in the monomial basis and their coefficients, respectively. By using the DC PSE method, the spatial derivatives $Q^\beta$ up to second order are given as:

$$Q^{1,0} \equiv \frac{\partial}{\partial x}, \quad Q^{0,1} \equiv \frac{\partial}{\partial y} \tag{31}$$

and

$$Q^{1,1} \equiv \frac{\partial^2}{\partial x \partial y} = \frac{\partial^2}{\partial y \partial x}, \quad Q^{2,0} \equiv \frac{\partial^2}{\partial x^2}, \quad Q^{0,2} \equiv \frac{\partial^2}{\partial y^2} \tag{32}$$

### 2.4. Vorticity Boundary Conditions

In finite difference methods, a number of formulae can be used to impose the vorticity boundary conditions [33,39]. These formulae, generally referred to as local, compute vorticity values on a given node on the boundary from the vorticity values in the interior domain, without involving other nodes on the boundary. The most widely used method is Thom's formula [39]. Unfortunately, applying these formulae in real applications had limited success. As stated in [33], vorticity boundary conditions should be global, in the sense that computing the boundary vorticity values should involve vorticity values at nodes in the interior and the boundary. Herein, to achieve global vorticity boundary conditions, we impose vorticity boundary conditions through strong form meshless differential operators, used to compute spatial derivatives. This way, imposing vorticity boundary conditions is consistent with the rest of the algorithm (described in Section 2.2) and becomes quite straightforward.

DC PSE methods are recognized as one of the most accurate and efficient numerical methods to compute derivatives on an irregularly distributed set (cloud) of nodes [34,35,38]. For the stream function-vorticity N-S Equations, given the velocity field values computed previously by the updated stream function values $\psi^{n+1}$ (Step 2, Section 2.2), we can compute the updated vorticity values $\omega^{n+1}$ for the entire spatial domain (including boundaries) using the strong form meshless operators for first order spatial derivative as

$$\omega^{n+1} \;=\; \frac{\partial v^{n+1}}{\partial x} - \frac{\partial u^{n+1}}{\partial y} \tag{33}$$

We have already computed the updated values $\omega^{n+1}$ at the interior nodes (Step 1 in the Euler explicit solver). The updated vorticity values on the boundary nodes are computed using the updated velocity values and the DC PSE operators (described in Section 2.3) for spatial derivatives (Equation (27)).

### 2.5. Poisson Solver

To compute the stream function field values (Equation (2)), we numerically solve a Poisson-type equation. Depending on the flow problem and spatial discretization, a direct or iterative solver is chosen accordingly. Iterative and direct solvers for elliptic type PDEs are well established. Direct solvers are robust and easy to use but can be computationally costly and memory intensive when the systems of equations to be solved are very large (e.g., a few million degrees of freedom). On the other hand, a decisive advantage of iterative solvers is their low memory usage, which is significantly less than a direct solver for the same sized problems. Unfortunately, iterative solvers are less robust than direct solvers and may fail to compute the numerical solution.

It is computationally more efficient to use iterative solvers to solve the Poisson equation when a large number of nodes is used. There are many iterative solvers for Poisson type equations. For example, Gauss-Seidel or successive over relaxation (SOR) algorithms have been widely used. These solvers are accurate but relatively computationally demanding because they need $O\left(N^2\right)$ iterations to converge ($N$ being the total number of grid points). Multi-grid algorithms have been developed to accelerate convergence and the computational effort has been significantly reduced [40]. For equally-spaced grids, fast Fourier transform (FFT) solvers are the fastest available algorithms for solving Poisson equations [41].

In the present study, we use a direct solver. The discrete Laplace operator defined in the Poisson type equation is accurately discretized using DC PSE method, which performs accurately on uniform and locally refined Cartesian grids. The grid is fixed in time so a LU factorization can be precomputed, which reduces the computational cost significantly. For up to 4 million nodes, the factorization is fast and requires low memory. For larger number of nodes, we utilize iterative solvers, such as conjugate gradient (CG) and biconjugate gradient stabilized (BiCGSTAB), since the linear systems are always positive definite, diagonally dominant and symmetric.

*2.6. Critical Time Step*

To compute the critical time step needed to obtain stable results, we rewrite Equation (1) as

$$\omega^{n+1} = \omega^n + dt\left(\frac{\partial \psi^n}{\partial x}\frac{\partial \omega^n}{\partial y} - \frac{\partial \psi^n}{\partial y}\frac{\partial \omega^n}{\partial x} + \frac{1}{Re}\nabla^2\omega^n\right) \tag{34}$$

which can be written in matrix form as

$$\boldsymbol{\omega}^{n+1} = [\mathbf{I} + \delta t\mathbf{A}]\boldsymbol{\omega}^n \tag{35}$$

where $\boldsymbol{\omega}^{n+1}$ is a column vector of dimension $N \times 1$ ($N$ is the number of nodes). The $N \times N$ matrix $\mathbf{A}$ is defined as the DC PSE approximation of the convection and diffusion terms, viz.

$$\mathbf{A} = \boldsymbol{K} + \boldsymbol{L} = \frac{\partial \psi^n}{\partial x}\frac{\partial \boldsymbol{\omega}^n}{\partial y} - \frac{\partial \psi^n}{\partial y}\frac{\partial \boldsymbol{\omega}^n}{\partial x} + \frac{1}{Re}\nabla^2\boldsymbol{\omega}^n. \tag{36}$$

where

$$\boldsymbol{K} = \text{diag}(\boldsymbol{Q}^{1,0}\boldsymbol{\psi}^n)\cdot\boldsymbol{Q}^{0,1} - \text{diag}(\boldsymbol{Q}^{0,1}\boldsymbol{\psi}^n)\cdot\boldsymbol{Q}^{1,0} \tag{37}$$

and

$$\boldsymbol{L} = \frac{1}{Re}(\boldsymbol{Q}^{2,0} + \boldsymbol{Q}^{0,2}) \tag{38}$$

with $\text{diag}(\boldsymbol{Q}^{1,0}\boldsymbol{\psi}^n)$ and $\text{diag}(\boldsymbol{Q}^{1,0}\boldsymbol{\psi}^n)$ being diagonal matrices of dimension $N \times N$. The explicit method defined by Equation (34) is stable if

$$|1 + \delta t\lambda_A| \leq 1 \tag{39}$$

where $\lambda_A$ are the eigenvalues of the matrix $\mathbf{A}$. The above stability condition requires the eigenvalues $\lambda_A$ to be either real or negative, leading to

$$\delta t \leq \frac{2}{|\lambda_A|} \tag{40}$$

or complex with negative real parts, leading to

$$\delta t \leq \frac{2Re(\lambda_A)}{|\lambda_A|^2} \tag{41}$$

When the problem discretization includes a large number of nodes, matrix $\mathbf{A}$ becomes very large, and calculating the eigenvalues is not practical. The terms of matrix $\mathbf{A}$ are determined by the Laplacian operator $\boldsymbol{L}$, which consists of second order derivatives, and by the advection operator $\boldsymbol{K}$, which includes only first order derivatives. The relative weight of the two operators on the structure of matrix $\mathbf{A}$ is dictated by discretisation and velocity field values (velocity is related to stream function and vector potential field values); a more refined discretization leads to a higher weight of operator $\boldsymbol{L}$ (diffusion) and lower influence of operator $\boldsymbol{K}$ (convection), while higher Reynolds number and higher velocity field values leads to a higher influence of operator $\boldsymbol{K}$. When the nodal discretization is not adequately refined, the higher weight of the operator $\boldsymbol{K}$ can lead to eigenvalues with a positive real part; in such case explicit time integration is not possible and discretization has to be refined.

For refined discretizations, the resulting matrix is diagonally dominant, having its eigenvalues distributed close to the real axis. We can use the Gershgorin circle theorem [31] to estimate the maximum time step using Equations (36) and (37) with an upper bound of the maximum eigenvalue of matrix $\mathbf{A}$. Given the composition of matrix $\mathbf{A}$, the upper bound of the maximum eigenvalue can be computed without assembling the matrix:

$$|\lambda_A| \leq \max_i \left( \sum_j \left( |L_{ij}| + |K_{ij}| \right) \right) \tag{42}$$

Equation (42) clearly shows that the eigenvalues of matrix **A** depend on the spatial derivatives of the field variables, computed using the DC PSE method. Therefore, how derivatives are computed reflects on the critical time step. We notice that for the same Reynolds number and the same nodal discretization, the critical time step increases when spatial derivatives are computed using a large number of nodes (more than 40) in the support domain of each node, compared with the critical time step computed using a small number of nodes (around 15). In some cases, this increase in the critical time step can be up to two orders of magnitude, which decreases the computational cost dramatically.

The reason for the increased critical time step is the upwind inherent nature of meshless methods. Meshless methods are locally based methods and use neighboring nodes to compute spatial derivatives. Furthermore, since matrix **A** involves derivatives of the stream function, by computing the critical time step using the Gershgorin circle theorem we ensure that the computation resembles the estimation of the critical time step based on the nodal spacing $\left( \propto \Delta x^{-1} \right)$ and the Courant–Friedrichs–Lewy (CFL) condition $\left( \propto u \mathrm{Re} \Delta x^{-1} \right)$. Therefore, by adjusting the number of support domain nodes, we can adjust the critical time step to be comparable to that used by implicit time integration schemes.

## 3. Algorithm Verification

To demonstrate the efficiency and accuracy of the proposed numerical scheme, we first apply our method to well-established benchmark problems in computational fluid dynamics, including lid-driven cavity flow, the backward-facing step (BFS), and the unbounded flow past a cylinder.

For all the examples presented here, we examine the accuracy and the efficiency of the proposed scheme for each benchmark problem. Furthermore, we examine the dependence of the critical time step on the grid resolution, Reynolds number and nodes in the support domain. Computations were conducted using an Intel i7 quad core processor with 16 GB RAM.

### 3.1. Lid-Driven Cavity

Our first example involves incompressible, non-stationary flow in a square cavity (Figure 1). Despite its geometrical simplicity, the lid-driven cavity flow problem exhibits a complex flow regime, mainly due to the vortices formed in the center and at the corners (bottom left and right, and upper left) of the square domain. We impose no-slip boundary conditions at the bottom and vertical walls (left and right wall), while the top wall slides with unit velocity ($U = 1$), generating the interior viscous flow.

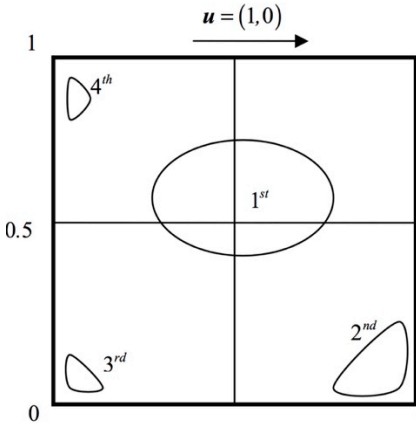

**Figure 1.** Geometry and boundary conditions for the lid-driven cavity flow problem.

In most numerical studies on the lid-driven cavity flow problem, the flow regime reaches a steady state solution for Reynolds (*Re*) numbers lower than *Re* = 10,000. In our study, we report on the results obtained by the proposed scheme for Reynolds numbers *Re* = 7500 and 10,000. We consider both uniform Cartesian and irregular nodal distributions. For uniform nodal distributions, our previous studies [27–29,35] show that for Reynolds numbers up to *Re* = 10,000, a uniform Cartesian grid with resolution of 361 × 361 captures the forming vortices and provides a grid-independent and accurate numerical solution. In the present study, to highlight the efficiency of the proposed scheme, we use successively finer uniform Cartesian grids, from 601 × 601 to 2048 × 2408 nodes. We notice that for *Re* = 10,000 the 1024 × 1024 grid resolution can capture all the flow scales and can be used for Direct Numerical Simulation (DNS) studies.

We use the Gershgorin circle theorem to define the critical time step for different nodal distributions (Tables 1 and 2) and for various Reynolds numbers. Table 1 lists the critical time step computed using the Gershgorin circle theorem for several grid resolutions and different Reynolds numbers (up to *Re* = 10,000). For the results reported in Table 1, 20 nodes were used in the support domain. We further examine the dependence of the critical time step on the number of neighboring nodes in the support domain. Table 2 lists the critical time step computed using the Gershgorin circle theorem with 40 nodes in the support domain. We can observe that the critical time step increases with the number of nodes in the support domain.

**Table 1.** Critical time step for various grid resolution and Reynolds (*Re*) numbers, using 20 nodes in the support domain.

| Grid Resolution | *Re* = 5000 | *Re* = 7500 | *Re* = 10,000 |
|---|---|---|---|
| 601 × 601 | $3.4 \times 10^{-3}$ | $5.1 \times 10^{-3}$ | $6.7 \times 10^{-3}$ |
| 1024 × 1024 | $1.1 \times 10^{-3}$ | $1.6 \times 10^{-3}$ | $2 \times 10^{-3}$ |
| 2048 × 2048 | $3.1 \times 10^{-4}$ | $4.4 \times 10^{-4}$ | $5 \times 10^{-4}$ |

**Table 2.** Critical time step for various grid resolution and *Re* numbers, using 40 nodes in the support domain.

| Grid Resolution | *Re* = 5000 | *Re* = 7500 | *Re* = 10,000 |
|---|---|---|---|
| 601 × 601 | $7.6 \times 10^{-3}$ | $8.2 \times 10^{-3}$ | $9.2 \times 10^{-3}$ |
| 1024 × 1024 | $3.1 \times 10^{-3}$ | $3.5 \times 10^{-3}$ | $3.7 \times 10^{-3}$ |
| 2048 × 2048 | $1.2 \times 10^{-3}$ | $1.3 \times 10^{-3}$ | $9.1 \times 10^{-4}$ |

Table 3 lists the computational time (in s) for computing the spatial derivatives for various grid resolutions, and for the numerical solution of N-S equations (for each time iteration) in the case of *Re* = 10,000. Recall that we precompute the LU factorization for the discrete Laplacian operator defined for the Poisson type equation for the stream function. Hence, steps 1 and 2 must be performed only once at the beginning of the simulation.

**Table 3.** Computational time (in seconds) for computing (**1**) spatial derivatives, (**2**) factorization and (**3**) numerical solution for various grid resolutions.

| Grid Resolution | 1. Derivatives | 2. Factorization | 3. Solution (in s)/Iteration |
|---|---|---|---|
| 601 × 601 | 3.92 s | 4.97 s | 0.22 s |
| 1024 × 1024 | 11.10 s | 21.35 s | 0.67 s |
| 2048 × 2048 | 56.79 s | 274.70 s | 41.39 s |

Figure 2 shows the *u*- velocity profiles along the vertical line passing through the geometric center of the cavity, and the *v*- velocity profiles along the horizontal line passing through the geometric center

of the cavity, for *Re* = 7500 and 10,000, obtained using the proposed scheme and compared to those reported by Ghia et al. [42], which are considered as benchmark solution.

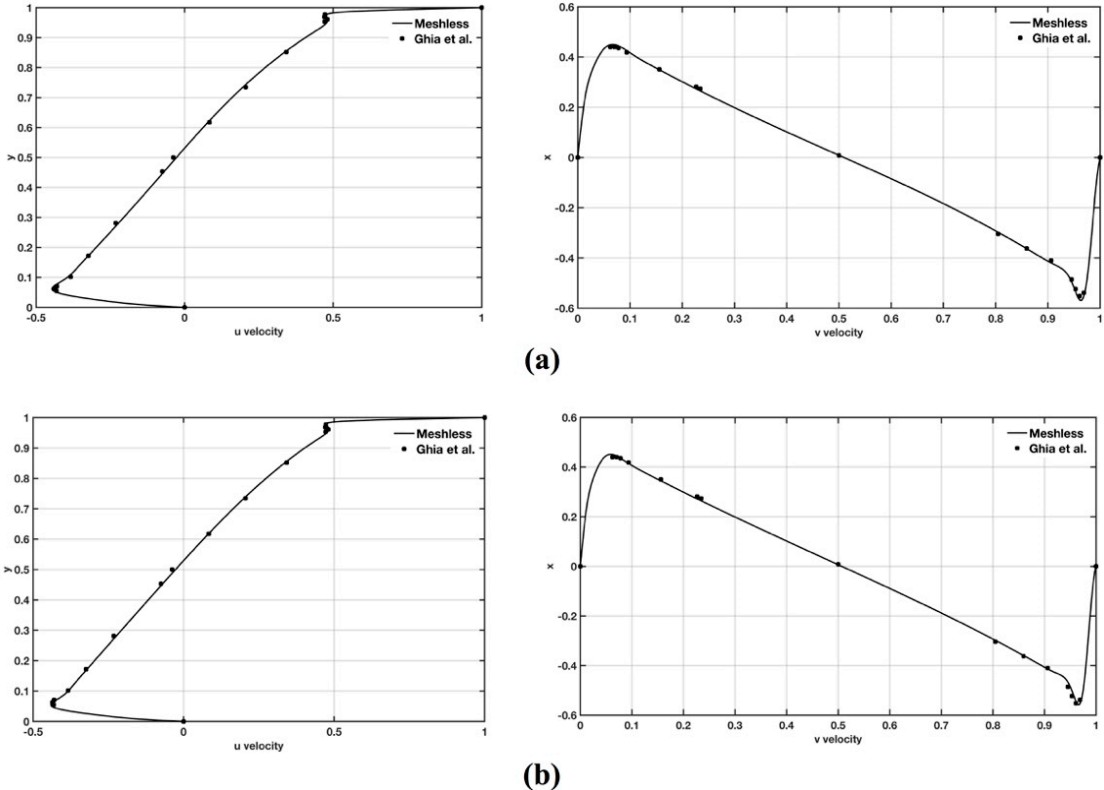

**Figure 2.** Velocity profiles along the vertical line passing through the geometric center of the cavity for *u*- velocity component (left) and horizontal line passing through the geometric center of the cavity for the *v*- velocity component (right) for (**a**) *Re* = 7500 and (**b**) *Re* = 10,000. The results we obtained in this study using our methods described in section Methods Governing Equations and Solution Procedure (indicated as "Meshless" in the velocity profiles) are compared with the results reported in Ghia et al. [42].

Table 3 demonstrates the efficiency of the proposes scheme and its ability to solve relatively large algebraic systems (Poisson equation for stream function) with low computational cost.

We set the Reynolds number to *Re* = 10,000 and the total time for the flow simulation to $T_{final}$ = 250. At that time, the flow regime has all the characteristic features of steady state [42]. We use a 1024 × 1024 grid resolution and a time step for the simulations of $dt = 5 \times 10^{-4}$.

Figures 3–5 show stream function and vorticity contours and demonstrate the formation of secondary vortices (Figures 3 and 4) which appear as the Reynolds number increases. There are often used as qualitative results, to highlight the accuracy of the numerical method. In detail, Figure 3 shows the stream function contour plots for *Re* = 7500, while Figure 4 shows the stream function contour plots for *Re* = 10,000. Magnified views of the secondary vortices are also included. The stream function values for these contours are listed in Table 4.

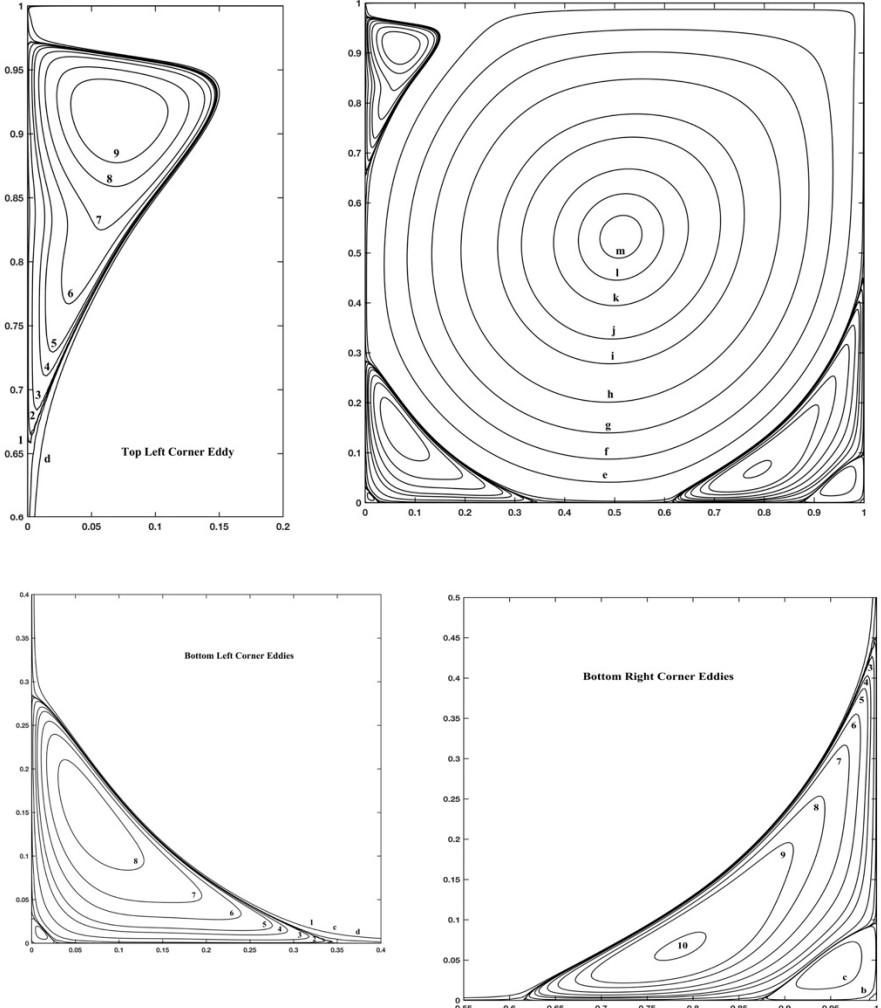

**Figure 3.** Streamline contours of primary, secondary and additional corner vortices for lid-driven cavity flow with *Re* = 7500.

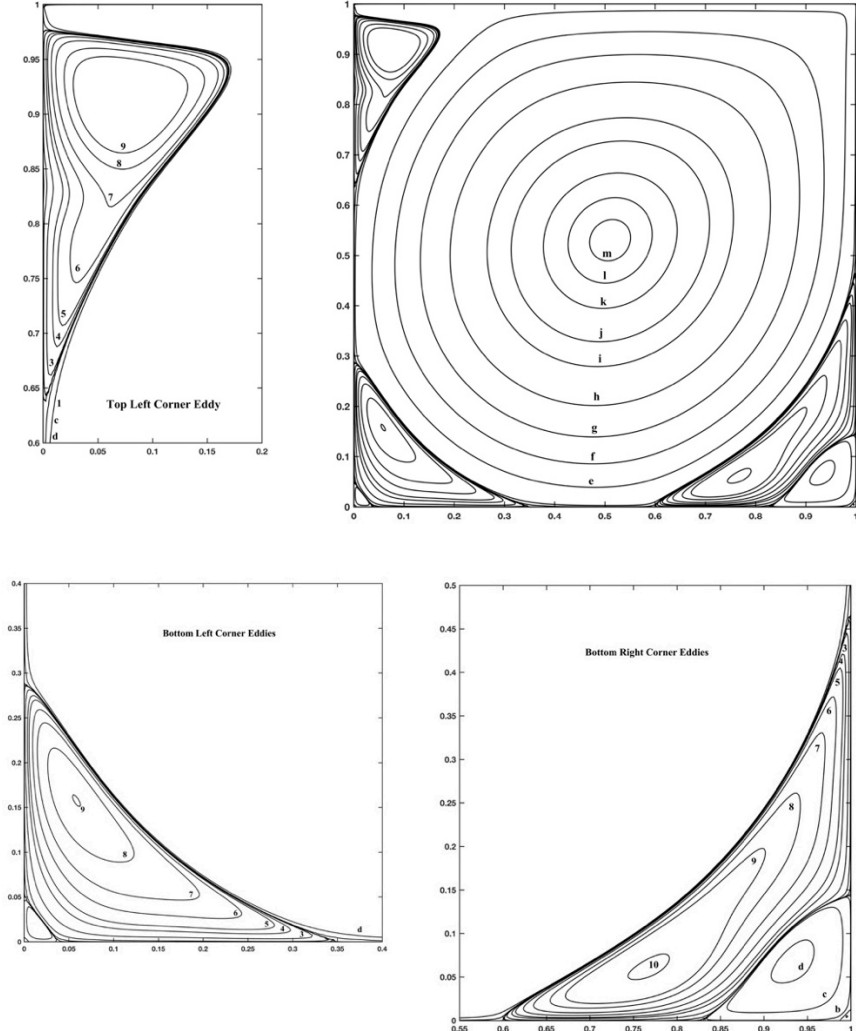

**Figure 4.** Streamline contours of primary, secondary and additional corner vortices for lid-driven cavity flow with *Re* = 10,000.

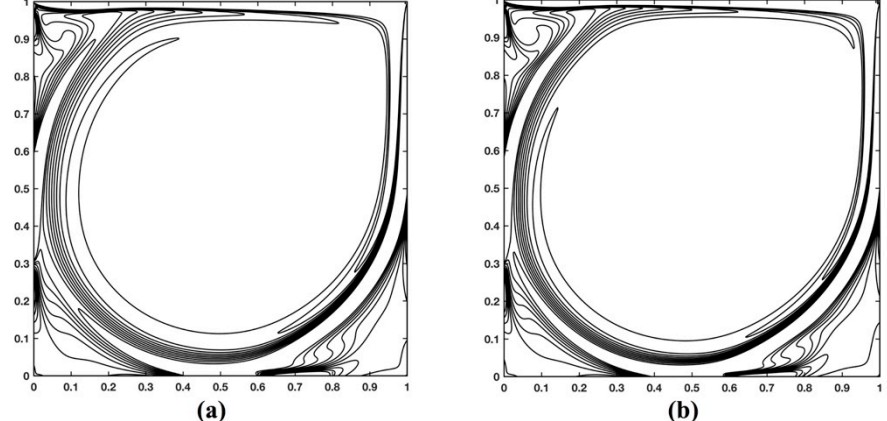

**Figure 5.** Vorticity isocontour patterns for (**a**) *Re* = 7500 and (**b**) *Re* = 10,000.

**Table 4.** Values for streamline and vorticity contours in Figures 3 and 4.

| | Stream Function | | Vorticity | | |
|---|---|---|---|---|---|
| Contour Level | Value of ψ | Contour Number | Value of ψ | Contour Number | Value of ψ |
| a | $-1.0 \times 10^{-10}$ | | | | |
| b | $-1.0 \times 10^{-7}$ | 0 | $1.0 \times 10^{-8}$ | | |
| c | $-1.0 \times 10^{-5}$ | 1 | $1.0 \times 10^{-7}$ | | |
| d | $-1.0 \times 10^{-4}$ | 2 | $1.0 \times 10^{-6}$ | 0 | 0 |
| e | $-0.0100$ | 3 | $1.0 \times 10^{-5}$ | $\pm 1$ | $\pm 0.5$ |
| f | $-0.0300$ | 4 | $5.0 \times 10^{-5}$ | $\pm 2$ | $\pm 1.0$ |
| g | $-0.0500$ | 5 | $1.0 \times 10^{-4}$ | $\pm 3$ | $\pm 2.0$ |
| h | $-0.0700$ | 6 | $2.5 \times 10^{-4}$ | $\pm 4$ | $\pm 3.0$ |
| i | $-0.0900$ | 7 | $5.0 \times 10^{-4}$ | 5 | 4.0 |
| j | $-0.1000$ | 8 | $1.0 \times 10^{-3}$ | 6 | 5.0 |
| k | $-0.1100$ | 9 | $1.5 \times 10^{-3}$ | | |
| l | $-0.1150$ | 10 | $3.0 \times 10^{-3}$ | | |
| m | $-0.1175$ | | | | |

Figure 5 shows the vorticity contours for *Re* = 7500 and 10,000. The vorticity values for the contours are listed in Table 4.

To verify the applicability of the proposed scheme in irregular nodal configurations, we compute the flow regime for *Re* = 10,000 using randomly distributed nodes. We use 496,987 nodes to discretize the spatial domain, which provides a grid-independent numerical solution (for the uniform Cartesian grid, a grid independent solution was obtained using a 401 × 401 grid resolution with 160,801 nodes). We obtain the irregular point cloud through a 2D triangular mesh generator (MESH2D-Delaunay-based unstructured mesh-generation). To compare the results with those obtained using the uniform nodal distribution (regular Cartesian grid), we interpolate the velocity, stream function and vorticity values computed using the irregular point cloud onto the uniform nodal distribution using the moving least squares (MLS) approximation method [43] and compute the maximum normalized root mean square error (NRMSE), defined as [44],

$$NRMSE = \frac{\sqrt{\frac{1}{N} \sum_{i=1}^{N} \left( u_i^{Cartesian} - u_i^{Irregular} \right)^2}}{u_{max}^{Irregular} - u_{min}^{Irregular}} \tag{43}$$

For all field variables considered, the NRMSE was less than $10^{-4}$, which highlights the accuracy of the proposed method for irregular nodal distributions.

We further demonstrate the accuracy of the proposed scheme, providing quantitative results. We use a uniform Cartesian grid of 1024 × 1024 resolution and we set the time step to $dt = 5 \times 10^{-4}$. In Table 5, we present the strengths and positions of the primary (located in the center of the cavity), secondary (bottom right corner) and ternary vortexes (bottom and top left corners). The numerical results are in excellent agreement with the results reported in Ghia et al. [42]. Table 6 lists the *u*-velocity values along vertical line through the center of cavity, and Table 7 the *v*- velocity values along horizontal line through the center of cavity.

**Table 5.** Comparison of center of primary, secondary, and ternary vortices, and the corresponding stream function values.

| *Re* | Reference | $\Psi_{min}$ | $(x_{1st}, y_{1st})$ | $\Psi_{max}$ | $(x_{2nd}, y_{2nd})$ | $\Psi_{min}$ | $(x_{3rd}, y_{3rd})$ | $\Psi_{max}$ | $(x_{4th}, y_{4th})$ |
|---|---|---|---|---|---|---|---|---|---|
| 7500 | Ghia [42] | −0.1199 | (0.5117, 0.5322) | $3.30 \times 10^{-3}$ | (0.7813, 0.0625) | $1.47 \times 10^{-3}$ | (0.0645, 0.1504) | $2.05 \times 10^{-3}$ | (0.0664, 0.9141) |
| | Present | −0.1132 | (0.5100,0.5333) | $3.12 \times 10^{-3}$ | (0.7783, 0.0634) | $1.36 \times 10^{-3}$ | (0.0612, 0.1554) | $1.97 \times 10^{-3}$ | (0.0614, 0.9267) |
| 10,000 | Ghia [42] | −0.1197 | (0.5117, 0.5333) | $3.42 \times 10^{-3}$ | (0.7656, 0.0586) | $1.52 \times 10^{-3}$ | (0.0586, 0.1641) | $2.42 \times 10^{-3}$ | (0.0703, 0.9141) |
| | Present | −0.1183 | (0.5109, 0.5331) | $3.36 \times 10^{-3}$ | (0.7642, 0.0582) | $1.51 \times 10^{-3}$ | (0.0583, 0.1643) | $2.41 \times 10^{-3}$ | (0.0703, 0.9142) |

**Table 6.** *u*- velocity along vertical line through the center of cavity.

| | Re = 7500 | | Re = 10,000 | |
|---|---|---|---|---|
| *y* | **Ghia [42]** | **Present** | **Ghia [42]** | **Present** |
| 1.00000 | 1.00000 | 1.00000 | 1.00000 | 1.00000 |
| 0.9766 | 0.47244 | 0.46956 | 0.47221 | 0.47244 |
| 0.9688 | 0.47048 | 0.46818 | 0.47783 | 0.47797 |
| 0.9609 | 0.47323 | 0.47186 | 0.48070 | 0.48034 |
| 0.9531 | 0.47167 | 0.47184 | 0.47804 | 0.47814 |
| 0.8516 | 0.34228 | 0.3381 | 0.34635 | 0.34685 |
| 0.7344 | 0.20591 | 0.19527 | 0.20673 | 0.20665 |
| 0.6172 | 0.08342 | 0.08142 | 0.08344 | 0.08332 |
| 0.5000 | −0.03800 | −0.03798 | 0.03111 | 0.03154 |
| 0.4531 | −0.07503 | −0.07245 | −0.07540 | −0.07534 |
| 0.2813 | −0.23176 | −0.23905 | −0.23186 | −0.23136 |
| 0.1719 | −0.32393 | −0.31801 | −0.32709 | −0.32729 |
| 0.1016 | −0.38324 | −0.38291 | −0.38000 | −0.38012 |
| 0.0703 | −0.43025 | −0.42988 | −0.41657 | −0.41648 |
| 0.0625 | −0.4359 | −0.43050 | −0.42537 | −0.42587 |
| 0.0547 | −0.43154 | −0.42935 | −0.42735 | −0.42775 |
| 0.0000 | 0.0000 | 0.0000 | 0.0000 | 0.0000 |

**Table 7.** *v*- velocity along horizontal line through the center of cavity.

| | Re = 7500 | | Re = 10,000 | |
|---|---|---|---|---|
| *x* | **Ghia [42]** | **Present** | **Ghia [42]** | **Present** |
| 1.00000 | 0.00000 | 0.00000 | 0.00000 | 0.00000 |
| 0.9688 | −0.53858 | −0.53889 | −0.54302 | −0.54367 |
| 0.9609 | −0.55216 | −0.55258 | −0.52487 | −0.52432 |
| 0.9531 | −0.52347 | −0.52363 | −0.49099 | −0.49056 |
| 0.9453 | −0.4859 | −0.48576 | −0.45863 | −0.45834 |
| 0.9063 | −0.41050 | −0.41082 | −0.41496 | −0.41439 |
| 0.8594 | −0.36213 | −0.36257 | −0.36737 | −0.36756 |
| 0.8047 | −0.30448 | −0.30432 | −0.30719 | −0.30739 |
| 0.5000 | 0.00824 | 0.00851 | 0.00831 | 0.00845 |
| 0.2344 | 0.27348 | 0.27378 | 0.27224 | 0.27267 |
| 0.2266 | 0.28117 | 0.28135 | 0.28003 | 0.28056 |
| 0.1563 | 0.3506 | 0.35082 | 0.35070 | 0.35078 |
| 0.0938 | 0.41824 | 0.41867 | 0.41487 | 0.41463 |
| 0.0781 | 0.43564 | 0.43591 | 0.43124 | 0.43165 |
| 0.0703 | 0.4403 | 0.44059 | 0.43733 | 0.43745 |
| 0.0625 | 0.43979 | 0.43962 | 0.43983 | 0.43953 |
| 0.0000 | 0.00000 | 0.00000 | 0.00000 | 0.00000 |

### 3.2. Backward-Facing Step

The second benchmark problem considered is the backward-facing step [45]. This problem has been studied by several researchers using different numerical methods and is considered to be a demanding flow problem to solve, mainly due to the flow separation that occurs when the fluid passes over a sharp corner and re-attaches downstream [1,45,46].

Figure 6 shows the spatial domain for the backward facing step. The coordinate system is centered at the step corner, with the *x*-coordinate being positive in the downstream direction, and the *y*-coordinate across the flow channel. The height *H* of the channel is set to *H* = 1 (ranging from (0, −0.5) to (0, 0.5)), while the step height and upstream inlet region are set to *H*/2. To ensure fully developed flow, the downstream channel length *L* is set to *L* = 30 *H*. The inlet velocity has a parabolic profile, with horizontal component $u(y) = 12y - 24y^2$ for $0 \leq y \leq 0.5$, which gives a maximum inflow velocity of

$u_{max} = 1.5$ and average velocity of $u_{avg} = 1$. At the outlet, we assume fully developed flow ($du/dx = 0$, $v = 0$). The Reynolds number is defined as $Re = u_{avg}H/v_f$, with $v_f$ being the kinematic viscosity.

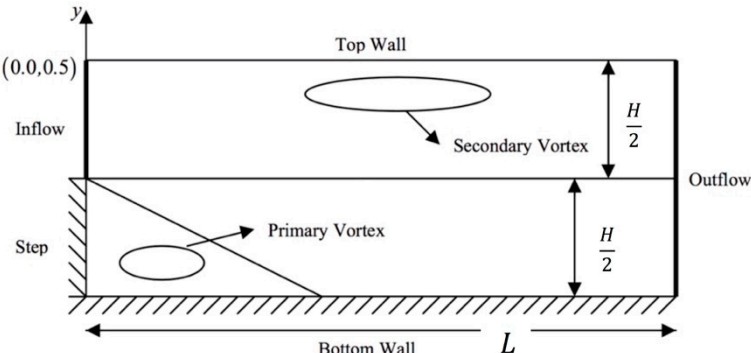

**Figure 6.** Geometry for the backward-facing step flow problem.

We first discretize the flow domain with a uniform Cartesian grid. In our previous studies [29,35], we have shown that grid with resolution $31 \times 901$ is sufficient to compute a grid-independent numerical solution for Reynolds numbers up to $Re = 800$. As in the previous section, to highlight the efficiency of our scheme, we use a denser grid with resolution $121 \times 3630$, resulting in 439,230 nodes. We set the total time $T_{total} = 250$ (to ensure that the solution will reach steady state), and the time step $dt = 10^{-4}$. The simulation terminates when the NRMSE of the time derivative of the stream function and vorticity field values in two successive time steps is less than $10^{-6}$. The time needed to create the grid was 0.023 s, and it takes 0.3 s to update the solution for each time step. We obtained a steady state solution after 100,000 time steps.

Figure 7 shows the stream function and vorticity contours for $Re = 800$. The flow separates at the step corner and vortices are formed downstream. We can observe the two vortices formed at the lower and upper wall. After reattachment of the upper wall eddy, the flow in the duct slowly recovers towards a fully developed flow. We compute the separation and reattachment points at $L_{lower} \approx 6.1$ for the lower wall separation zone, $L_{upper} \approx 5.11$ for the upper separation zone, where the separation begins at $x \approx 5.19$. Our numerical findings show good agreement with other numerical methods for 2D computations [1,46]. In [1], the authors used a finite difference method and predicted separation lengths of $L_{lower} \approx 6.0$ and $L_{upper} \approx 5.75$, while [46] using the A Fluid Dynamics Analysis Program (FIDAP) code (Fluid dynamics International Inc., Evanston, IL, USA) predicted $L_{lower} \approx 5.8$ and the upper $L_{upper} \approx 4.7$.

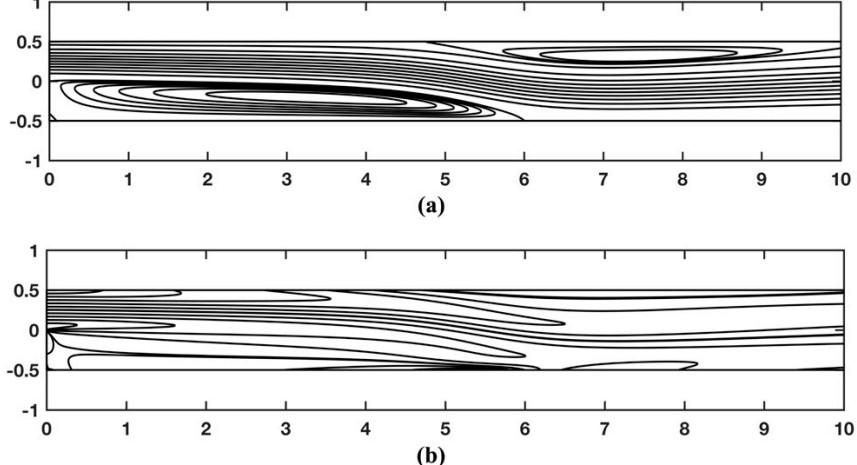

**Figure 7.** Contour plots of the (**a**) stream function and (**b**) vorticity for $Re = 800$ for the backward-facing step flow problem using the stream function and vorticity values reported in Gartling [45].

Figure 8 shows the comparisons of the *u*- and *v*- velocity components and vorticity values, between the present scheme and those obtained using the finite element method (FEM) [45], along the line $x = 7$ and $x = 15$ for $Re = 800$. Additionally, we computed the shear stress, defined as $\frac{\partial u}{\partial y}$ on the lower and upper wall, for increasingly refined nodal resolution. Figure 9a shows plots of the shear stress for the upper and lower wall, while Figure 9b displays the convergence of the upper wall shear stress when successively denser point clouds are used. We can see that shear stress converges and is accurate when compared to results of [45]. The accurate computation of spatial derivatives is a particular advantage of the strong form meshless method. In contrast, convergence of many finite element methods can be obtained only in terms of integral norms, and the point-wise convergence for the velocity gradient cannot be guaranteed even in the case of a smooth numerical solution [47].

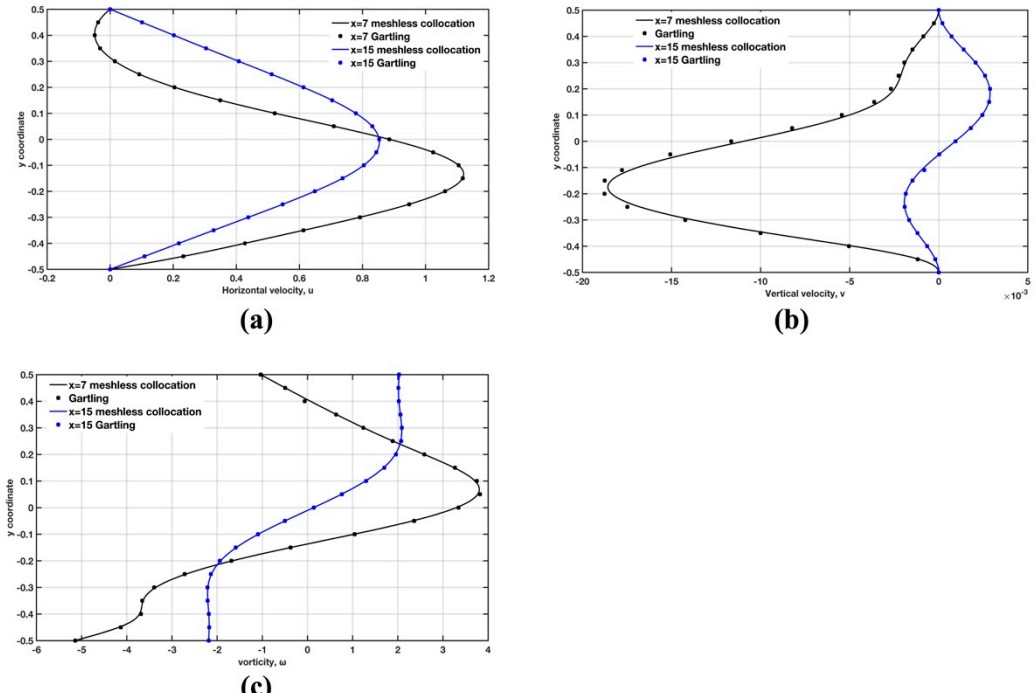

**Figure 8.** (**a**) Horizontal and (**b**) vertical velocity profiles, and (**c**) vorticity profiles at $x = 7$ and $x = 15$ for the backward-facing step flow problem with $Re = 800$.

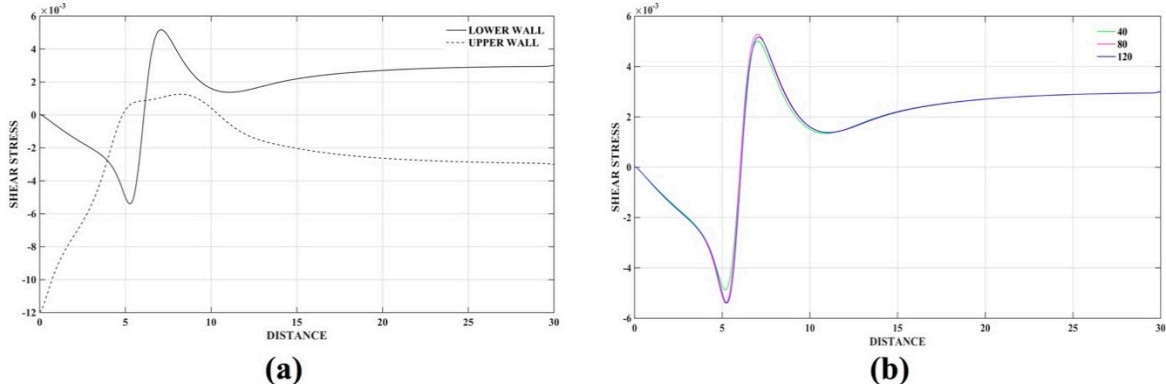

**Figure 9.** (**a**) Shear stress for the upper and lower wall for $Re = 800$ for the backward-facing step flow problem (**b**) convergence of the upper wall shear stress with successively denser point clouds (40, 80 and 120 correspond to the number of nodes in the *y*-direction).

Tables 8 and 9 list the velocity and vorticity field values at the at $x = 7$ and $x = 15$, respectively, for $Re = 800$.

**Table 8.** Cross-sectional profile at $x = 7$.

| $y$ | $u$ | | $v\left(\times 10^{-2}\right)$ | | $\omega$ | |
|---|---|---|---|---|---|---|
| | Gartling [45] | Present | Gartling [45] | Present | Gartling [45] | Present |
| 0.50 | 0.000 | 0.000 | 0.000 | 0.000 | −1.034 | −1.033 |
| 0.45 | −0.038 | −0.038 | −0.027 | −0.027 | −0.493 | −0.491 |
| 0.40 | −0.049 | −0.049 | −0.086 | −0.085 | 0.061 | 0.065 |
| 0.35 | −0.032 | −0.031 | −0.147 | −0.144 | 0.635 | 0.641 |
| 0.30 | 0.015 | 0.016 | −0.193 | −0.186 | 1.237 | 1.245 |
| 0.25 | 0.092 | 0.093 | −0.225 | −0.213 | 1.888 | 1.894 |
| 0.20 | 0.204 | 0.205 | −0.268 | 0.249 | 2.588 | 2.585 |
| 0.15 | 0.349 | 0.349 | −0.362 | −0.333 | 3.267 | 3.246 |
| 0.10 | 0.522 | 0.523 | −0.544 | −0.502 | 3.751 | 3.709 |
| 0.05 | 0.709 | 0.709 | −0.823 | −0.767 | 3.821 | 3.767 |
| 0.00 | 0.885 | 0.885 | −1.165 | −1.095 | 3.345 | 3.295 |
| −0.05 | 1.024 | 1.024 | −1.507 | −1.425 | 2.362 | 2.328 |
| −0.10 | 1.105 | 1.104 | −1.778 | −1.691 | 1.046 | 1.034 |
| −0.15 | 1.118 | 1.117 | −1.925 | −1.838 | −0.374 | −0.367 |
| −0.20 | 1.062 | 1.061 | −1.917 | −1.836 | −1.684 | −1.661 |
| −0.25 | 0.948 | 0.947 | −1.748 | −1.678 | −2.719 | −2.687 |
| −0.30 | 0.792 | 0.792 | −1.423 | −1.378 | −3.392 | −3.355 |
| −0.35 | 0.613 | 0.613 | −1.000 | −0.960 | −3.658 | −3.636 |
| −0.4 | 0.428 | 0.427 | −0.504 | −0.482 | −3.687 | −3.719 |
| −0.45 | 0.232 | 0.232 | −0.118 | −0.113 | −4.132 | −4.177 |
| −0.50 | 0.000 | 0.000 | 0.000 | 0.000 | −5.140 | −5.146 |

**Table 9.** Cross-sectional profile at $x = 15$.

| $y$ | $u$ | | $v\left(\times 10^{-2}\right)$ | | $\omega$ | |
|---|---|---|---|---|---|---|
| | Gartling [45] | Present | Gartling [45] | Present | Gartling [45] | Present |
| 0.50 | 0.000 | 0.000 | 0.000 | 0.000 | 2.027 | 2.033 |
| 0.45 | 0.101 | 0.101 | 0.021 | 0.021 | 2.013 | 2.018 |
| 0.40 | 0.202 | 0.202 | 0.072 | 0.072 | 2.023 | 2.027 |
| 0.35 | 0.304 | 0.304 | 0.140 | 0.139 | 2.058 | 2.058 |
| 0.30 | 0.408 | 0.408 | 0.207 | 0.206 | 2.090 | 2.085 |
| 0.25 | 0.512 | 0.512 | 0.260 | 0.259 | 2.075 | 2.063 |
| 0.20 | 0.613 | 0.613 | 0.288 | 0.286 | 1.959 | 1.943 |
| 0.15 | 0.704 | 0.704 | 0.283 | 0.281 | 1.703 | 1.687 |
| 0.10 | 0.779 | 0.779 | 0.245 | 0.243 | 1.298 | 1.283 |
| 0.05 | 0.831 | 0.831 | 0.180 | 0.177 | 0.761 | 0.751 |
| 0.00 | 0.853 | 0.853 | 0.095 | 0.093 | 0.141 | 0.137 |
| −0.05 | 0.844 | 0.844 | 0.003 | 0.002 | −0.500 | −0.447 |
| −0.10 | 0.804 | 0.804 | −0.081 | −0.081 | −1.096 | −1.086 |
| −0.15 | 0.737 | 0.737 | −0.147 | −0.147 | −1.588 | −1.574 |
| −0.20 | 0.649 | 0.649 | −0.185 | −0.185 | −1.939 | −1.923 |
| −0.25 | 0.547 | 0.547 | −0.191 | −0.191 | −2.139 | −2.125 |
| −0.30 | 0.438 | 0.438 | −0.166 | −0.166 | −2.213 | −2.203 |
| −0.35 | 0.328 | 0.328 | −0.119 | −0.119 | −2.210 | −2.206 |
| −0.40 | 0.218 | 0.218 | −0.065 | −0.065 | −2.184 | −2.185 |
| −0.45 | 0.109 | 0.109 | −0.019 | −0.019 | −2.174 | 2.177 |
| −0.50 | 0.000 | 0.000 | 0.000 | 0.000 | −2.185 | −2.189 |

### 3.3. Unbounded Flow Past a Cylinder

We examine the case of external flow past a circular cylinder in an unbounded domain [48]. The flow domain is large compared to the dimensions of the cylinder, as shown in Figure 10. The cylinder cross-section has a radius $R_c = 0.5$ and is located at the origin $O$ (0,0) of a square domain with dimensions $-10 \le x \le 30$ and $-20 \le y \le 20$.

The uniform flow is not perturbed near the inlet, which is far from the body. Therefore, we assume that the inflow velocity $u_{inlet}$ behaves like the potential flow $u_{potential}$ given as:

$$\boldsymbol{u}_{potential} = \left( U_\infty \left( 1 - \frac{R_c^2}{x^2 + y^2} + \frac{2R_c^2 y^2}{(x^2 + y^2)^2} \right), U_\infty \frac{-2R_c^2 xy}{(x^2 + y^2)^2} \right) \tag{44}$$

or in terms of the stream function:

$$\psi_{potential} = U_\infty y \left( 1 - \frac{R_c^2}{x^2 + y^2} \right) \tag{45}$$

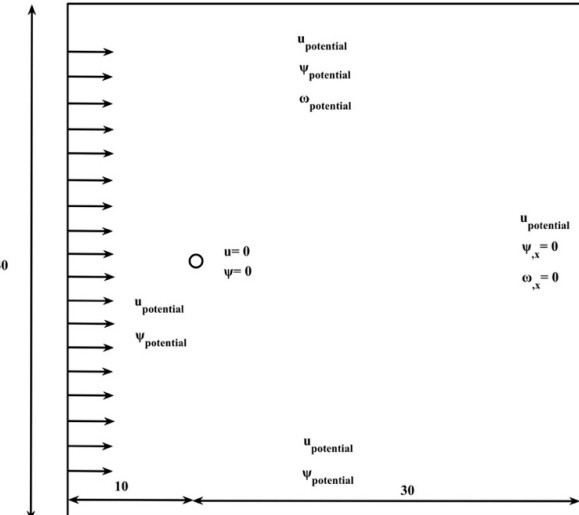

**Figure 10.** Geometry and boundary conditions for flow past a cylinder.

The boundary conditions for the stream function $\psi$ and vorticity $\omega$ on the remaining boundaries (the top and bottom wall, outlet and cylinder surface) are:

- At the inlet, top and bottom walls ($B$ = inlet, top and bottom)

$$\boldsymbol{u}_B = \left. \boldsymbol{u}_{potential} \right|_B$$

$$\psi_B = \left. \psi_{potential} \right|_B$$

- At the outlet

$$\frac{\partial \psi}{\partial x} = 0$$

$$\frac{\partial \omega}{\partial x} = 0$$

- On the cylindrical surface

$$\psi_{cylinder} = 0$$

$$\boldsymbol{u}_{cylinder} = 0$$

We represent the flow domain with a uniform Cartesian embedded grid, locally refined in the vicinity of the cylinder (Figure 11a). We use 402,068 nodes (0.5 s to create the nodal distribution) and we set Reynolds number to $Re = 40$. We use a time step of $dt = 10^{-4}$ for the entire flow simulation. It takes 0.3 s to update the solution for each time step, and we obtained a steady state solution after 20,000 time steps. The spatial derivatives (up to 2nd order) are computed in 4.24 s using a C++ code. Additionally, we solve the flow problem using an irregular nodal distribution with 221,211 nodes, locally refined in the vicinity of the cylinder.

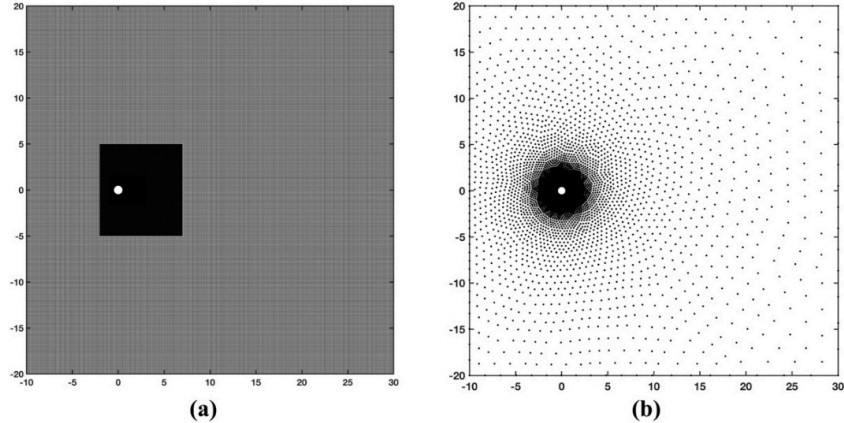

**Figure 11.** (**a**) Locally refined Cartesian and (**b**) irregular nodal configurations for the flow past cylinder flow problem.

We compute the following flow parameters: the pressure coefficient ($C_p$) on the body surface, the length ($L$) of the wake behind the body, the separation angle ($\theta_s$), and the drag coefficient ($C_D$) of the body. The drag and lift coefficient of the body are given by:

$$C_D = \frac{2F_b \cdot \hat{e}_x}{U_\infty^2 D} \tag{46}$$

$$C_L = \frac{2F_b \cdot \hat{e}_y}{U_\infty^2 D} \tag{47}$$

where $D$ is the characteristic length of the body, $\hat{e}_x$ and $\hat{e}_y$ are the unit normal vector in $x$- and $y$-direction, respectively, and $F_b$ is the total drag force acting on the body. The total drag force is given as:

$$F_b = F_p + F_f \tag{48}$$

where $F_p$, the pressure drag, can be determined from the flux of vorticity on the surface of the cylinder as:

$$F_p = -R_C \int_0^{2\pi} v_f \frac{\partial \omega}{\partial r} \hat{e}_\theta d\theta \tag{49}$$

The friction drag $F_f$ may be computed from the vorticity on the surface of the body as:

$$F_f = R_C \int_0^{2\pi} v_f \omega \hat{e}_\theta d\theta \tag{50}$$

with $\hat{e}_\theta = -\sin \theta i + \cos \theta j$. Table 5 lists the recirculation length ($L_{rec}$) of the wake behind the body, the separation angle ($\theta_s$), and the drag coefficient ($C_D$) for $Re = 40$.

The numerical findings obtained by the proposed scheme are listed in Table 10, and are in good agreement with those in the literature [48–51]. The stream function and vorticity contours around the cylinder are illustrated in Figure 12 (as in [48]).

**Table 10.** Comparison of the wake length ($L_{sep}$), the separation angle ($\theta_{sep}$), and the drag coefficient ($C_D$) for Reynolds number $Re = 40$.

| Re | Reference | $L_{sep}$ | $\theta_{sep}$ | $C_D$ |
|----|-----------|-----------|----------------|-------|
|    | Dennis and Chang [49] | 2.345 | 53.8 | 1.522 |
|    | Fornberg [50] | 2.24 | N/A | 1.498 |
| 40 | Ding et al. [51] | 2.20 | 53.5 | 1.713 |
|    | Kim et al. [48] | 2.187 | 55.1 | 1.640 |
|    | Present | 2.30 | 53 | 1.542 |

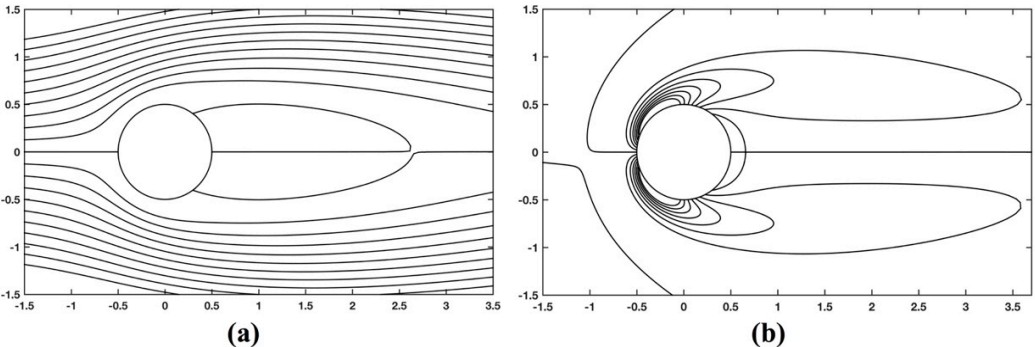

**Figure 12.** (**a**) Stream function and (**b**) vorticity contours around the body for flow behind a cylinder using a Cartesian embedded nodal distribution.

## 4. Numerical Results

In this section, we examine the applicability and reproducibility of the proposed scheme for several flow cases involving complex geometries.

### 4.1. Vortex Shedding Behind Cylinders

In the previous example (unbounded flow past a cylinder), we considered the flow past a cylinder in an unbounded domain. In this example, we study the vortex shedding behind a cylinder in a rectangular duct [52] (internal flow), as shown in Figure 13. The duct has length $L$ = 2.2 m and height $H$ = 0.41 m, while the cylinder is located at point $O$ (0.2, 0.2) and has diameter $D$ = 0.1 m. The kinematic viscosity of the fluid is set to $v_f$ = 0.001 m$^2$/s. The Reynolds number is defined as $Re = U_m D/v_f$, with the mean velocity $U_m$ given as $U_m(x,y,t) = 2U(0,H/2,t)/3$. The inflow velocity ($x$- direction component) is set to $U(0, y) = 4U_m y\left(\frac{H-y}{H^2}\right)$, and for $U_m$ = 1.5 m/s yields a Reynolds number $Re$ = 100. At the outlet, we consider fully developed flow ($du/dx = 0$), while at the remaining boundaries we apply no-slip boundary conditions. The total time for the simulation was set to $T_{tot}$ = 8 s. The critical time step is computed and monitored for each time step (the calculation involves stream function and vorticity values) to ensure that CFL conditions are fulfilled.

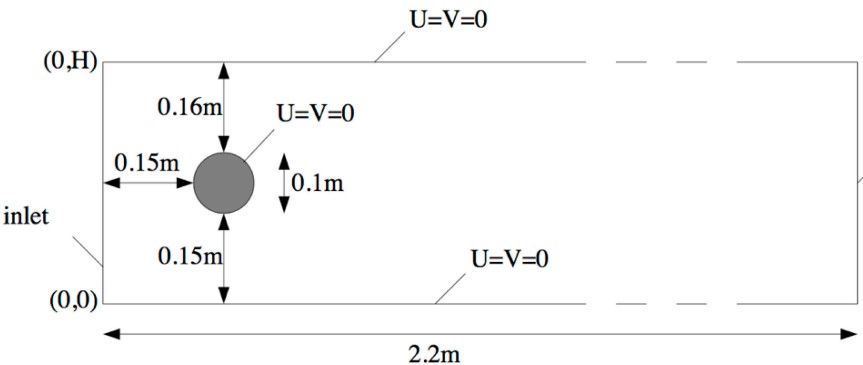

**Figure 13.** Spatial domain for flow behind a cylinder.

First, we consider a uniform Cartesian embedded grid with grid spacing $h$, defined by the average distance of the boundary nodes on the cylinder circumference. We conducted a grid independence analysis, applying successively refined Cartesian grids, starting from 34,999 (90 nodes on the circular boundary) up to 406,678 (720 nodes on the circular boundary) nodes.

The time needed to create the grid of 406,678 nodes is 0.8 s. We delete the nodes located close to the boundary (with distance less than 0.25 $h$), since they would increase the condition number of the Vandermonde matrix. A Vandermonde matrix with too large condition number would affect the accuracy of the spatial derivative computation and consequently the overall precision of the numerical

simulation. A grid independent solution is obtained with 203,890 nodes (360 nodes on the circular boundary). The critical time step computed for this grid resolution is $\delta t_{critical} = 2 \times 10^{-3}$. In the simulations, we set the time step to $dt = 10^{-3}$ s. For each time step, it takes 0.38 s to update the solution.

Figure 14a shows the stream function contours at time instances $t = 2, 4$ and 8 s, while Figure 14b plots the vorticity isocontours. Figure 15 plots the drag $C_D = \frac{2F_b \cdot \hat{e}_x}{U_m^2 D}$ and lift $C_L = \frac{2F_b \cdot \hat{e}_y}{U_m^2 D}$ coefficients, computed with our meshless explicit scheme (as in the previous example). To highlight the applicability of our code for locally refined nodal distributions, we use a uniform Cartesian embedded grid, locally refined in the vicinity of the cylinder. As before, the average distance of the cylinder nodes dictates the nodal spacing $h_d$ of the Cartesian grid in the refined part of the domain. Nodes of the coarse grid have spacing $h_c = 2h$, and nodes located close to the cylinder (with distance less than 0.25 $h_d$) are deleted since they affect the condition number of the Vandermonde matrix and decrease the accuracy of the numerical results. The flow domain is represented by 321,897 nodes, which ensure a grid independent solution. For the simulations, we set the time step to $dt = 10^{-3}$ s.

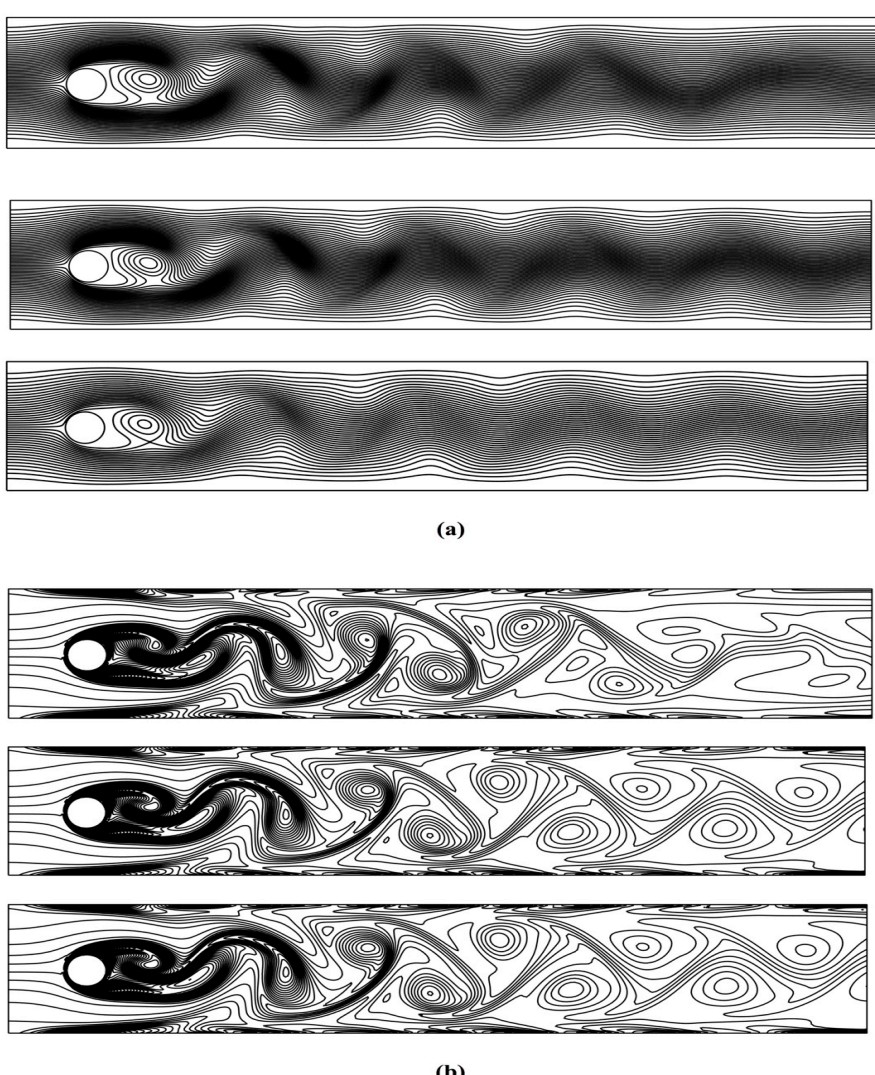

(a)

(b)

**Figure 14.** (**a**) Stream function and (**b**) vorticity contours for flow around a cylinder at time $t = 2, 4$ and 8 s.

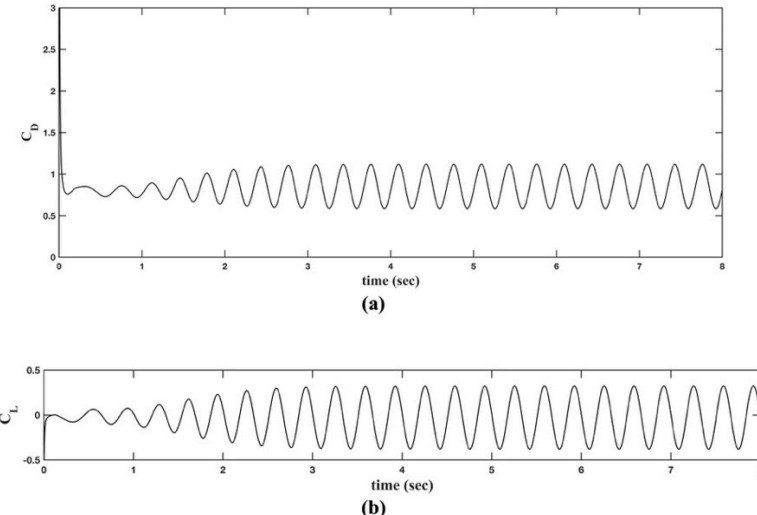

**Figure 15.** (**a**) Drag and (**b**) lift coefficient in time computed by the proposed scheme.

Additionally, to highlight the versatility of the proposed scheme, we examined the flow in the rectangular duct with multiple (seven in total) cylindrical obstacles. The length of the duct was increased to $L = 4.4$ m in order to ensure fully developed flow at the outlet. The boundary conditions applied are the same as before. We use Cartesian embedded and irregular nodal distributions (Figure 16) to represent the flow domain.

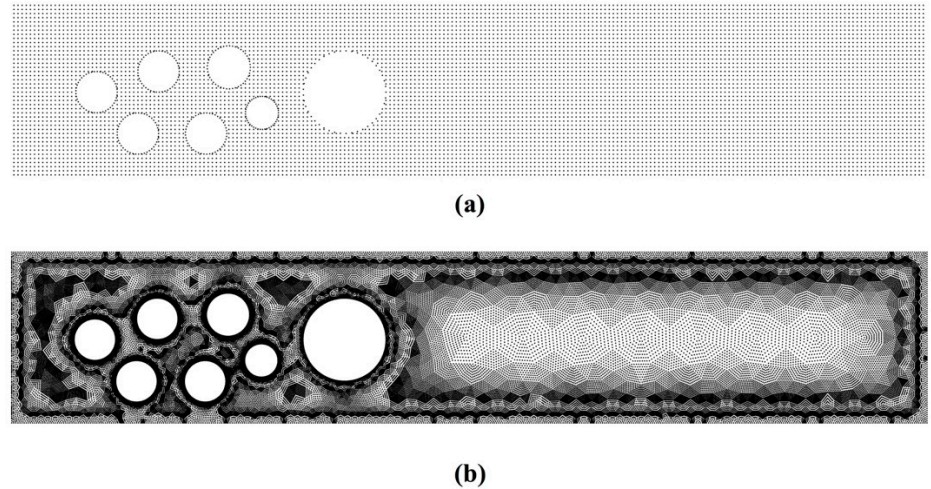

**Figure 16.** (**a**) Cartesian embedded and (**b**) irregular nodal distributions used in duct flow problems considering multiple cylindrical obstacles.

We conducted a comprehensive analysis in order to obtain a grid independent numerical solution. For the Cartesian grid we use 279,178 nodes (0.42 s to create the nodal distribution), while for the irregular point cloud we use 353,617 nodes (1.2 s to create the irregular nodal distribution). Both nodal distributions provided a grid independent numerical solution. The total time was set to $T_{tot} = 8$ s, the time step was set to $dt = 10^{-4}$ s. For each time step, the time needed to update the solution was 0.32 s. Figure 17 displays the stream function and vorticity contours for $Re = 100$ at time $t = 1$ and $t = 2$ s.

We further demonstrate the accuracy of the proposed scheme, by comparing our results with those computed using the finite element method (FEM). Flow equations were solved using the Incremental Pressure Correction Scheme (IPCS), which is an operator splitting method [53]. For the meshless solver, we use 353,617 nodes, generated using a triangular mesh generator, and we set the time step to $dt = 10^{-3}$ s. For the finite element model, we solve the flow equations using the generated mesh and

FEniCS (https://fenicsproject.org). We compute the Normalized Root Mean Square Error (NRMSE), as defined in Equation (43), for both velocity components ($u_x$, $u_y$), considering as gold standard the velocity field values calculated using FEniCS. Figure 18 shows the NRMSE for the velocity components for the time interval between $t = 0$ and $t = 1$ s.

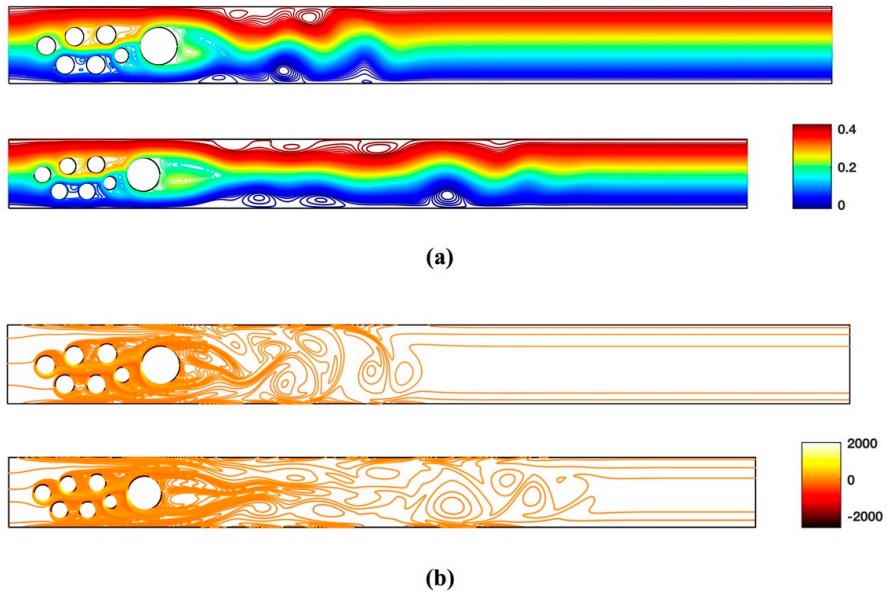

**Figure 17.** (**a**) Stream function and (**b**) vorticity contours for flow behind multiple cylinders at time $t = 1$ and $t = 2$ s.

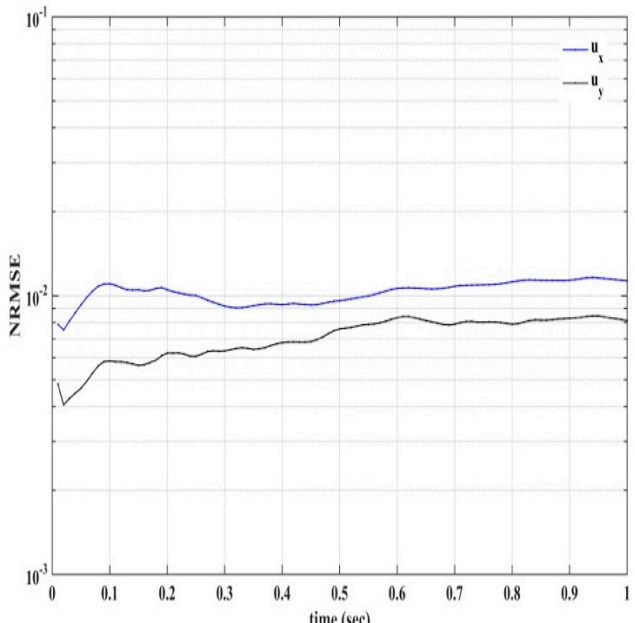

**Figure 18.** Normalized Root Mean Square Error (NRMSE) for the velocity components between the proposed scheme and the finite element solution.

### 4.2. Flow in a Duct with Irregular Geometry

Finally, we consider two flow cases in the flow domains shown in Figure 19. The first one has an irregularly shaped obstacle downstream, while in the second the flow domain is split into two branches, forming a bifurcation.

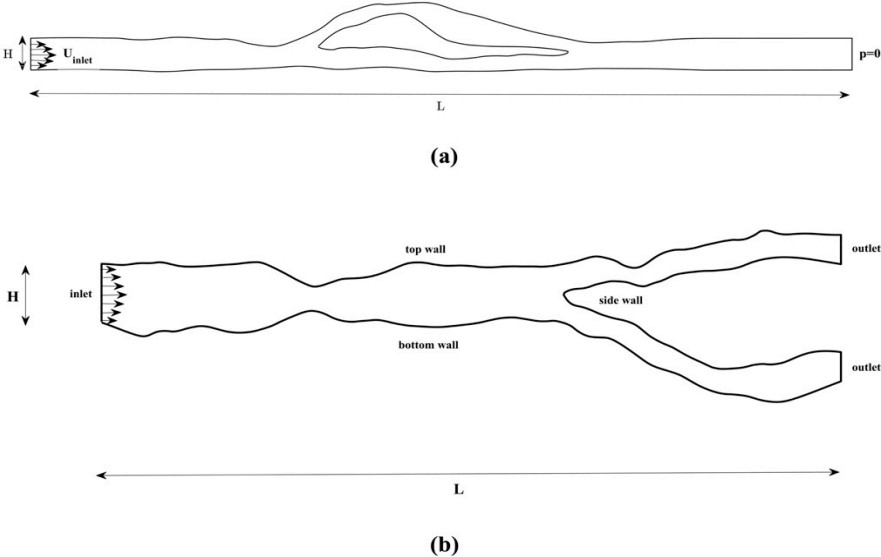

**Figure 19.** Geometry for the flow in the (**a**) bypass and (**b**) bifurcation domain.

The length *L* is set to *L* = 0.12 and 0.041 *m* for the first (Figure 19a) and second (Figure 19b) flow domain, respectively. In both test cases, the height of the duct is set to *H* = 0.0041 m.

The dynamic viscosity is defined as $\mu$ = 0.00032 kg/m s, and the fluid density is $\rho$ = 1050 kg/m$^3$. The inflow condition $U_{inlet} = 4U_m y\left(\frac{H-y}{H^2}\right)$, with $U_m$ = 0.035 m/s. The Reynolds number $Re = \rho U_m H / \mu$, with *D* being a characteristic length defined separately for each case. At the outlet, the flow is fully developed ($du/dx$ = 0). For the remaining boundaries we applied no-slip conditions. In both cases, we represent the flow domain with uniform Cartesian embedded grid, and irregular nodal distribution (Figure 20).

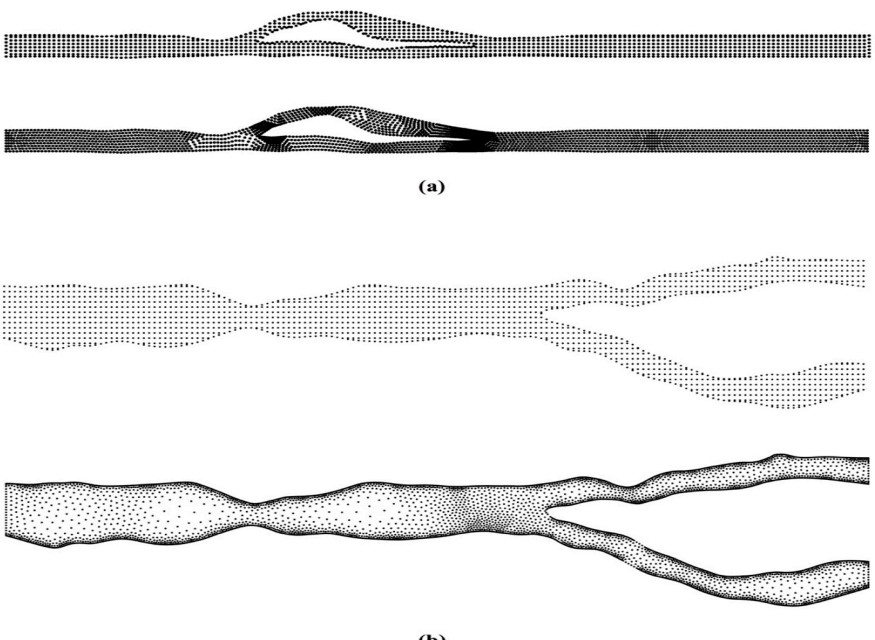

**Figure 20.** (**a**) Cartesian embedded and (**b**) irregular nodal distribution for the complex flow geometries.

First, we represent the flow domain with a uniform Cartesian embedded grid. For the irregularly shaped obstacle (bypass), we consider successively denser nodal distributions to obtain a grid independent numerical solution. We start with a grid spacing of *h* = 1/8200, *h* = 1/12,300 and

$h = 1/16{,}400$ for the coarse, moderate and dense Cartesian embedded grids, resulting in 51,468, 114,387 and 201,468 nodes, respectively (for the denser grid (201,468 nodes) it takes 0.25 s to create the nodal distribution). The total time for the simulation was set to $T_{tot} = 30$ s and the time step used was $dt = 10^{-4}$ s (for each time step the solution is computed in 0.35 s). Figure 21 shows the iso-contours for the stream function and vorticity field functions, at different time instances: $t = 0.5$ and $t = 1$ s, and for steady state (steady state is reached when, for two successive time instances $t^{n+1}$ and $t^n$ for both vorticity and stream function field values, $NRMSE/dt < 10^{-8}$).

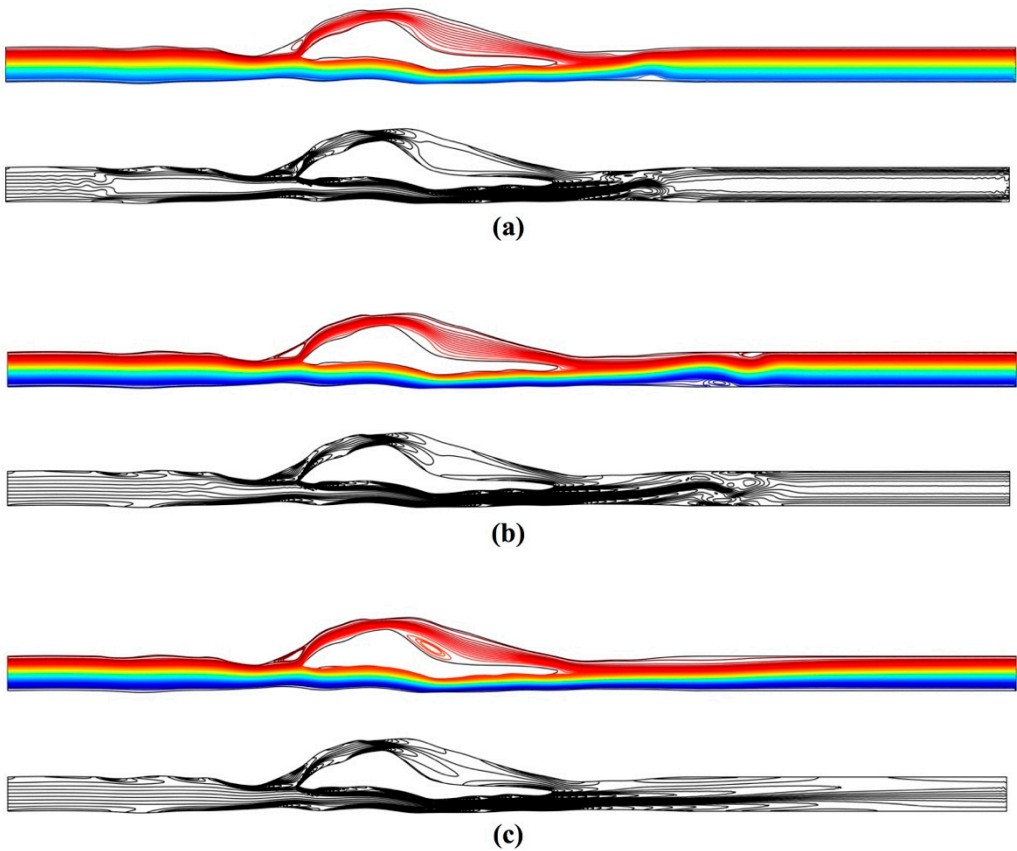

**Figure 21.** Stream function and vorticity contours for the bypass test case at (**a**) $t = 0.5$ s and (**b**) $t = 1$ s and (**c**) steady state.

We observe that two vortices are formed; the first is located at the inlet of the bypass, the second at the upper domain of the bypass which is moving towards the outlet of the flow domain. Figure 22a plots the $u$- velocity profile, computed at $x = 0.07$ m and $x = 0.08$ m at time $t = 3$ s for the coarse, moderate and dense nodal distributions, while Figure 22b plots the $u$- velocity profile, computed at $x = 0.07$ m and $x = 0.08$ m at time $t = 3$ s for the denser grid used.

Additionally, to highlight the applicability of the method to irregular nodal distributions, we solved the flow equations by representing the flow domain with nodes generated by a 2D triangular mesh generator (we use only the coordinates and not their connectivity). We used 392,122 nodes, which is higher than the number of nodes used in the Cartesian embedded grid. The numerical results obtained were compared against those obtained by the Cartesian grid and they were in excellent agreement.

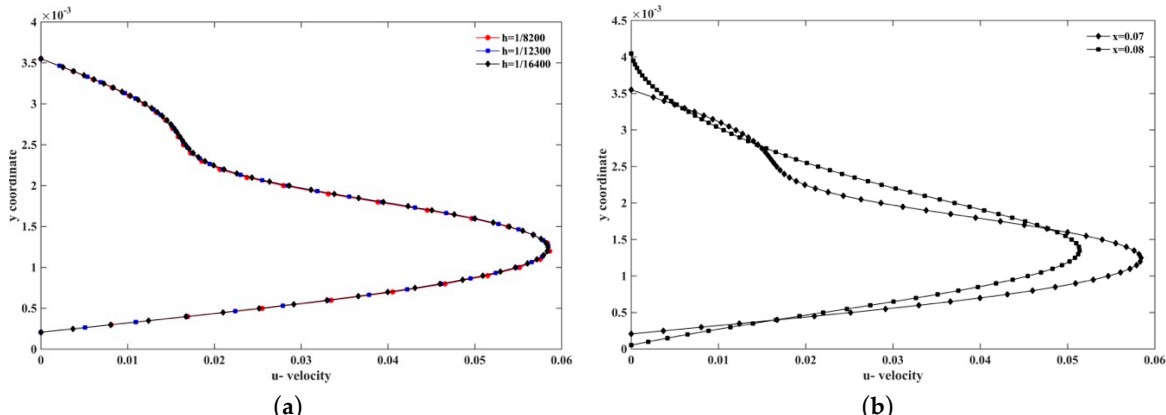

**Figure 22.** (**a**) *u*-velocity profiles at *x* = 0.07 at time *t* = 3 s for the coarse, moderate and dense nodal distributions for *Re* = 474 (*U_m* = 0.035 m/s) and (**b**) *u*-velocity profiles at *x* = 0.07 and 0.08 for flow in the bypass geometry.

For the bifurcated geometry, we use both uniform Cartesian and irregular nodal distributions. In the case of Cartesian embedded grid, we obtain a grid independent numerical solution by using successively denser nodal distributions, with the denser one consisting of 678,967 nodes (for the fine grid it takes 0.68 s to create the nodal distribution). The total time for the simulation was set to $T_{tot} = 30$ s and the time step used is $dt = 10^{-4}$ s, which is smaller than the critical time that ensures stability for the explicit solver (the solution in each time step is computed in 0.52 s). The inflow velocity $U_{inlet} = 4U_m y\left(\frac{H-y}{H^2}\right)$, with $U_m = 0.015$ m/s results in Reynolds number *Re* = 212. Figure 23 shows the iso-contours for the stream function and vorticity field functions, at time *t* = 0.5 and *t* = 1 s and for steady state (steady state is reached when, for two successive time instances $t^{n+1}$ and $t^n$ for both vorticity and stream function field values, $NRMSE/dt < 10^{-8}$).

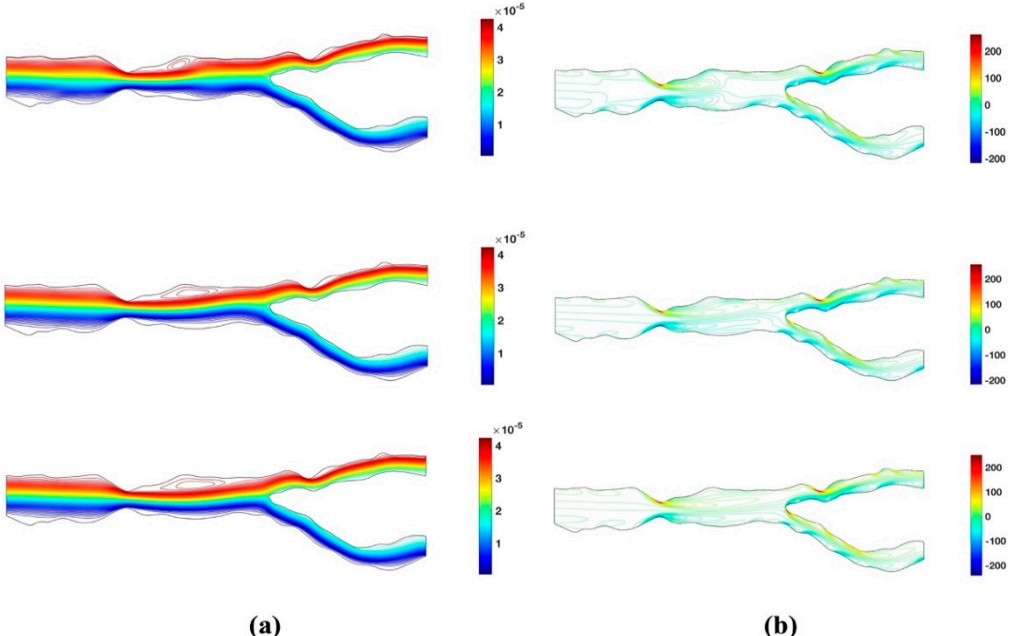

**Figure 23.** (**a**) Stream function and (**b**) vorticity contours for the bifurcation test case.

To further demonstrate the accuracy of the proposed scheme, we compare our numerical findings for both flow cases (bypass and bifurcation geometry) with those obtained using the finite element method. The flow domain is discretized using 392,122 and 678,967 nodes, respectively, and Taylor-Hood elements (as in the vortex shedding behind cylinders example). We use the IPCS flow

solver to numerically solve the flow equations. We compare both numerical solutions, computing the NRMSE for the two velocity components ($u_x$ and $u_y$) on the vertices of the triangular elements (the same nodes are used in the meshless point collocation flow solver). Figure 24 shows the NRMSE for the velocity components (bypass and bifurcation flow examples) for the time interval between $t = 0$ and $t = 1$ s.

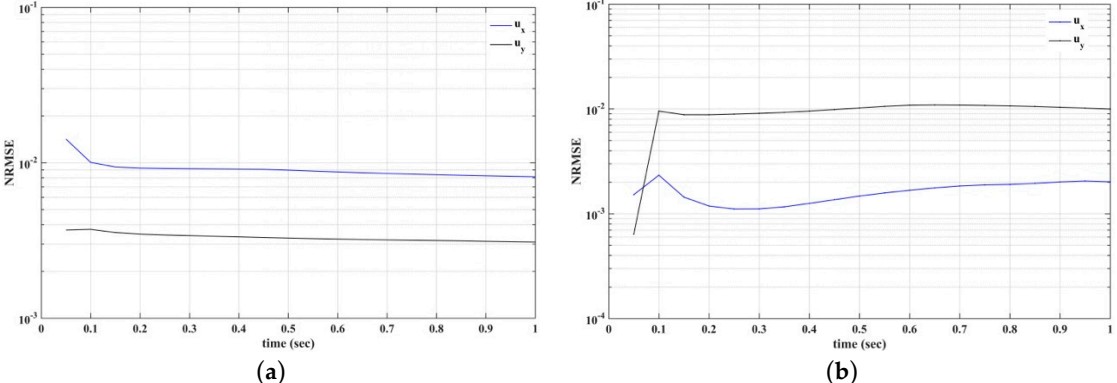

(a)　　　　　　　　　　　　　　　　　(b)

**Figure 24.** Normalized Root Mean Square Error (NRMSE) for the velocity components between the proposed scheme and the finite element solution for (**a**) bypass and (**b**) bifurcation flow cases.

## 5. Conclusions

In this contribution, we described a meshless point collocation algorithm for solving the stream function-vorticity formulation of N-S equations. We demonstrated our algorithm to work well in complex geometries like the ones shown in Section 4.1 (Vortex Shedding Behind Cylinders) and 4.2 (Flow in a Duct with Irregular Geometry). We demonstrate the accuracy of the proposed scheme by comparing our numerical results with those computed using the finite element method. An important advantage of our method is the ease and speed with which one can construct computational grids for flow domains with irregular shapes.

The meshless scheme based on DC PSE methods to compute spatial derivatives can be used in the case of Cartesian and Cartesian-embedded grids. The method works both efficiently and accurately for uniform Cartesian (embedded) grids and for irregular point clouds. We verified the accuracy of the proposed scheme through the following three benchmark problems: lid-driven cavity flow, backward-facing step and unbounded flow past a cylinder. The results were in good agreement with other published numerical studies.

Our proposed scheme offers several advantages over other commonly used methods:

- Rapid and easy generation of computational grids, as demonstrated by examples of the flow in vortex shedding behind cylinders and flow in a duct with irregular geometry in Sections 4.1 and 4.2, respectively.
- High accuracy, as demonstrated in verification examples discussed in Section 3.
- Computational efficiency, as demonstrated by time per iteration (Table 3) and total simulation time given for all examples in this paper. Our approach makes Direct Numerical Simulations (DNS) [54] possible (see lid driven cavity example with $1024 \times 1024$ grid resolution), even on personal computers.
- Critical time step is easily computed with low computational cost.
- Long critical time steps, comparable to those used in implicit schemes. The critical time step is related to the number of nodes located in the support domain and increases as the number of nodes increases (see Tables 1 and 2).
- The imposition of Dirichlet boundary conditions is straightforward.
- Easy and straightforward way to impose vorticity boundary conditions using spatial derivatives computed using the DC PSE method (Equation (33))

- Fast and accurate solution of the Poisson solver for the stream function equation (Equation (2)). The solution time/iteration (Table 3) is shorter compared to the solution time of other well-established numerical methods (see Figure 1 in [54]).
- Simplicity: we use a MATLAB code of ca. 150 lines to solve flow equations, and a C++ code of ca. 140 lines to compute spatial derivatives.

**Author Contributions:** Conceptualization, G.C.B.; Investigation, G.C.B.; Software, G.C.B.; Writing, G.C.B, B.F.Z. and K.M.; Writing—review and editing, G.C.B., B.F.Z., G.R.J., V.C.L., A.C.R.T, A.W. and K.M.

**Funding:** This research was supported in part by the Australian Government through the Australian Research Council's Discovery Projects funding scheme (project DP160100714).

**Acknowledgments:** This research was supported in part by the Australian Government through the Australian Research Council's Discovery Projects funding scheme (project DP160100714). The views expressed herein are those of the authors and are not necessarily those of the Australian Government or Australian Research Council.

**Conflicts of Interest:** The authors declare no conflict of interest.

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
