# Peer review of "An Explicit Meshless Point Collocation Solver for Incompressible Navier-Stokes Equations"

_fluids, doi:10.3390/fluids4030164_

Round 1
Reviewer 1 Report
In this paper, an explicit point collocation solver for 2 dimensional incompressible flows is developed. It is said the solver is for Navier-Stokes equations, but in fact it is just the stream function-vorticity formulation. At each time step, the Poisson Equation in which the left hand side is the Laplacian of the stream function while the right hand is negative omega that presents the vorticity is solved numerically. The point collocation method is employed for solving this Poisson Equation. Spatial derivatives needed for the time marching are computed with the scheme of Discretization Corrected Particle Strength Exchange (DC PSE). Followings are the reviewer’s suggestions.
1. The title should be corrected.
2. The Re shown up in Eq. (9) should be defined more clearly. Which Reynold’s number does it represent? The one at the n-th time step or the one at the (n+1)-th time step?
3. Eq. (36) might be incorrect. The very left hand side carries its physical meaning while the very right hand side is just the combination of linear operators. The middle part of this equation is in non-matrix form while the very left hand side and the very right hand side are in matrix forms.
4. The same error could be found in Eqs. (37) and (38).
5. Since the phenomena simulated are transient, or say time dependent, it would be more interesting if the focus is on the development of the flows. Showing the results at the steady state and mentioning how accurate the results are might not be enough.
6. The size of the time increment in the first test case should be mentioned.
7. In line 484, the “Re=800” should be expressed more clearly. Which flow speed and which length scale are used to determine this Reynolds’ number?
8. The title of the section beginning at line 678 might be incorrect.
9. The focus of present study is on 2D. All the equations and formulations are 2D. In the conclusion, showing the preliminary results of 3D flow simulation is inappropriate, unless the authors would like to add the 3D form of the governing equations, i.e. Eqs. (1) and (2) and give more detailed description on how this method could really work for 3D flows.
Reviewer 2 Report
This manuscript shows a new method of numerical solution for two-dimensional stream function-vorticity formulation of transient, incompressible, and viscous Navier-Stokes equations by meshless point collocation explicit solver. The numerical method was validated by simulating the lid-driven cavity flow, backward-facing step flow as well as the flow past a cylinder and the results agreed well with the existing studies. Also, the method was applied to the flows within a couple of complex structures.
This reviewer thinks that this study seems to contribute to further development of numerical simulations of fluid dynamics, while the manuscript needs to be revised based on the following comments and questions.
1. This reviewer believes that the objective of this manuscript is to propose a way of solving the N-S equations. Although the governing equations and numerical methods are well written, it is hard to find what is new in the manuscript. It would be much better if the authors more explicitly describe what was missing is the previous studies and how the authors overcome the problem.
2. The authors compare the computational time among different grid resolutions in Table 3. This reviewer believes that it is unclear what the authors intend to emphasize with this table. Is the “solution/iteration” for the present results very fast compared to the one by using the existing numerical methods, including FDM, FVM, and FEM?
3. Figures 2 - 5 present the results of the simulations of lid-cavity flow. However, the authors do not discuss these figures in the manuscript and they just explain what those are. Are the figures really needed to be shown? Also, this reviewer has the same question about Figure 12.
4. Considering the above question, this reviewer recommends that the authors would be better to present the results of one of the three benchmark simulations and add some more detailed information about more quantitative comparison between the present results and the results of existing results, such as numerical error, computational time, etc.
5. The simulations of “flow past a cylinder” and “vortex shedding behind cylinder” are almost the same condition except for the Reynolds number. Why did the authors separate these results (one is shown in section 3 while the other is in section 4)?
6. Regarding the figures 17 and 18, the authors did not discuss about these figures in the manuscript which is similar to figures 2-5. Why did the authors put these figures? They described in the manuscript (Line 659) “in order to obtain a grid independent numerical solution,” but they did not even mention whether or not the desired solution was obtained.
7. Regarding the simulations of flow in the complex geometries shown in figure 19, the reliability of the numerical results are not verified. This reviewer believes that the results can be verified by comparing them to the experimental results or the numerical results obtained by other numerical methods.
8. The results for the complex geometries are not summarized in the conclusions. What were the results shown for? This reviewer believes that the objective of showing these results are not well described in the manuscript, and thus the latter part of the manuscript does not seem to have a strength enough to support the authors’ new idea.
9. This reviewer recommends that the result of the three-dimensional lid-cavity flow will be presented in the next publication because this is just a preliminary result and does not include the data enough to validate the reliability of the simulation.
Minor comments
a. The font of figures are too small and better to be larger in the revised manuscript.
b. Table 5 seems to miss to inform the result of the present study.
c. The section number of 3.2 (Page 20) should be wrong.
d. Title of the section 3.2 (Page 23) should be different. Also, the section number should be different.
Round 2
Reviewer 1 Report
The authors have revised their manuscript following the reviewer’s suggestions. But some further revision is need. These are the comments of the reviewer.
The title would be better if it is “An explicit meshless point collocation solver for 2D incompressible flow using the stream function-vorticity formulation”. Indeed the governing equations in this paper are not Navier-Stokes equations, though they come from some derivations of the Navier-Stokes equations. The left hand side in Eq. 8 is a scalar while the right hand side is in vector form. It should be mentioned that this is because the flow is 2D. On the right hand side of Eq. 36, omega should be in bold font. Showing the NRMSE for the time interval between t=0 and t=1 is good, but what the reviewer suggested was not this. What the reviewer wanted to see is the snapshots like Figs. 3 to 5 but not at time instants before the steady state. It would help readers understand the development of the flow. Analogous stories could be told in the other test cases. The symbol D in Fig. 6 is not defined. Maybe it is L and L in this same figure might be H. There might be a typo in line 491.Author Response
Please see the attachment.

Reviewer 2 Report
The authors kindly responded to this reviewer's comments and manuscript has been revised based on the comments. This reviewer believes that the manuscript is now deserved to be published in Fluids, while this reviewer recommends the following minor revisions:
1. The font size of the axis and labels in some of the figures are better to be larger for readers.
2. The font size are better to be unified at least within a figure; for example, the font size of figure 8 (a) seems to be larger than those in figure 8 (b) and (c).
